# Signals from the niche promote distinct modes of translation initiation to control stem cell differentiation and renewal in the *Drosophila* testis

Ruoxu Wang[1], Mykola Roiuk[2,3], Freya Storer[4¤], Aurelio A. Teleman[2,3], Marc Amoyel[1,4]*

**1** Department of Cell and Developmental Biology, University College London, London, United Kingdom, **2** Signal Transduction in Cancer and Metabolism, German Cancer Research Center (DKFZ), Heidelberg, Germany, **3** Heidelberg University, Heidelberg, Germany, **4** School of Cellular and Molecular Medicine, University of Bristol, Bristol, United Kingdom

¤ Current address: Department of Life Sciences, Imperial College London, London, UK

* marc.amoyel@ucl.ac.uk

## Abstract

Stem cells have the unique ability among adult cells to give rise to cells of different identities. To do so, they must change gene expression in response to environmental signals. Much work has focused on how transcription is regulated to achieve these changes; however, in many cell types, transcripts and proteins correlate poorly, indicating that post-transcriptional regulation is important. To assess how translational control can influence stem cell fate, we use the *Drosophila* testis as a model. The testis niche secretes a ligand to activate the Janus kinase (JAK)/signal transducer and activator of transcription (STAT) pathway in two stem cell populations, germline stem cells (GSCs) and somatic cyst stem cells (CySCs). We find that global translation rates are high in CySCs and decrease during differentiation, and that JAK/STAT signaling regulates translation. To determine how translation was regulated, we knocked down translation initiation factors and found that the cap binding complex, eIF4F, is dispensable in differentiating cells, but is specifically required in CySCs for self-renewal, acting downstream of JAK/STAT activity. Moreover, we identify eIF3d1 as a key regulator of CySC fate, and show that two eIF3d1 residues subject to regulation by phosphorylation are critical to maintain CySC self-renewal. We further show that Casein Kinase II (CkII), which controls eIF3d1 phosphorylation, influences the binding of eIF3d and eIF4F in mammalian cells, and that CkII expression is sufficient to restore CySC function in the absence of JAK/STAT. We propose a model in which niche signals regulate a specific translation programme in which only some mRNAs are translated. The mechanism we identify allows stem cells to switch between modes of translation, adding a layer of regulation on top of transcription and providing cells with the ability to rapidly change gene expression upon receiving external stimuli.

**Data availability statement:** All relevant data are within the paper and its Supporting information files.

**Funding:** This work was supported by MRC Transition Award to MA – MR/W029219/1 and MRC Career Development Fellowship to MA – MR/P009646/2 https://www.ukri.org/councils/mrc/. The funders played no role in the study design, data collection and analysis or any other aspect of this manuscript.

**Competing interests:** The authors have declared that no competing interests exist.

**Abbreviations:** BDSC, Bloomington Drosophila Stock Center; CK2, Casein Kinase 2; CkII, Casein Kinase II; CySCs, cyst stem cells; DAP5, Death-Associated Protein 5; Dcp-1, Death caspase-1; EdU, 5-ethynyl-2′-deoxyuridine; eIF, eukaryotic initiation factor; Eya, Eyes absent; GSCs, germline stem cells; HCR, hybridization chain reaction; JAK, Janus kinase; NIG, Kyoto National Institute of Genetics; OPP, O-propargyl-puromycin; SSCT, saline sodium citrate solution; STAT, signal transducer and activator of transcription; TOP, terminal oligo-pyrimidine; Tor, Target of Rapamycin; Upd, Unpaired; VDRC, Vienna Drosophila Resource Center; zfh1, *Zn-finger homeodomain 1*; *4E-BP*, eIF4E-binding protein.

## Introduction

To maintain adult tissue homeostasis, stem cells must have the ability to respond to the needs of the tissue in a timely manner to produce differentiating offspring. Observations across tissues suggest that a pool of stem cells exists in a poised, or licensed state, enabling them to rapidly transition to a differentiated state upon signals from the environment [1–3]. Thus, the ability to switch gene expression profiles at short notice is critical to maintaining tissue integrity. How this is achieved is still poorly understood. Much work has focused on how transcription is regulated in stem and differentiating cells by signals from the niche, or supportive environment that maintains stem cell self-renewal, leading to an increasing understanding of gene regulatory networks and transcriptional landscapes during differentiation [4–8]. However, comparison of the transcriptome and translatome has revealed that protein expression is poorly correlated with transcription in many stem cells and their differentiated progeny [9–13]. Therefore, to fully understand how cell identity transitions occur, it is critical to study the post-transcriptional control of gene expression and its regulation by niche signals promoting self-renewal.

An important point of control of gene expression is the translation of mRNA. Two broad classes of mechanisms affecting mRNA translation in stem cells have been described: in the first, which is best characterized in the germline stem cells (GSCs) of the *Drosophila* ovary, sequence-specific RNA-binding proteins sequester and inhibit translation of mRNAs encoding differentiation factors [14,15]. The second class involves more global effects on translation rates which change during differentiation; surprisingly, these changes in global translation have major roles in determining cell fate, and act by preferentially affecting the translation of specific mRNAs, although in ways that are still poorly characterized [16,17]. Thus, different translational programmes are overlaid onto the transcriptional programmes of stem cells and their differentiating progeny, leading to a complex regulation of gene expression. Importantly, it remains unknown how niche signals, which direct the transcriptional programme of stem cell gene expression, impact global translation to promote stem cell-specific translational programmes.

To understand how changes in global translation rates can control cell fate, we focused on translation initiation, which is thought to be the rate-limiting step of protein synthesis and depends on several eukaryotic initiation factor (eIF) complexes [18]. Canonically, translation is initiated by binding of eIF4E to the 5′ m⁷G cap of the mRNA, where the eIF4F complex, composed of eIF4A, 4E and 4G subunits is assembled. eIF4F acts to recruit the eIF3 complex, a large multi-subunit initiation factor, which is bound to the ribosome and other eIFs, forming the 43S ribosome (Fig 1A). This results in recruitment of the 43S to the 5′ end of the mRNA, where it begins to scan for a translation start site. There is increasing evidence; however, that eIFs are not merely scaffolds, but that they are regulated and can selectively drive translation of specific transcripts. For instance, the *Drosophila* genome encodes eight eIF4E and two eIF4G paralogues [19]. Some of these are specifically expressed in the male germline and loss-of-function of these paralogues results in spermatogenesis defects at distinct stages [20–23], although it is still unclear whether this is due to specific functions of these factors or to different expression patterns or localization. Non-canonical translation can occur upon sequestration of eIF4E by eIF4E-binding protein (4E-BP), or expression of a variant eIF4G, eIF4G2, also known as Death-Associated Protein 5 (DAP5), which lacks the eIF4E binding domain. In many cases, cap-independent translation is thought to occur through internal ribosome entry sites. Of note, the eIF3 subunit, eIF3d, can bind the cap in the absence of eIF4F and promote translation of mRNAs depending on the structure of the 5′ untranslated region [24,25]. Indeed, it is thought that eIF3d drives cap-dependent translation together with DAP5 [26,27]. Cap-binding activity of eIF3d is dependent on phosphorylation by Casein

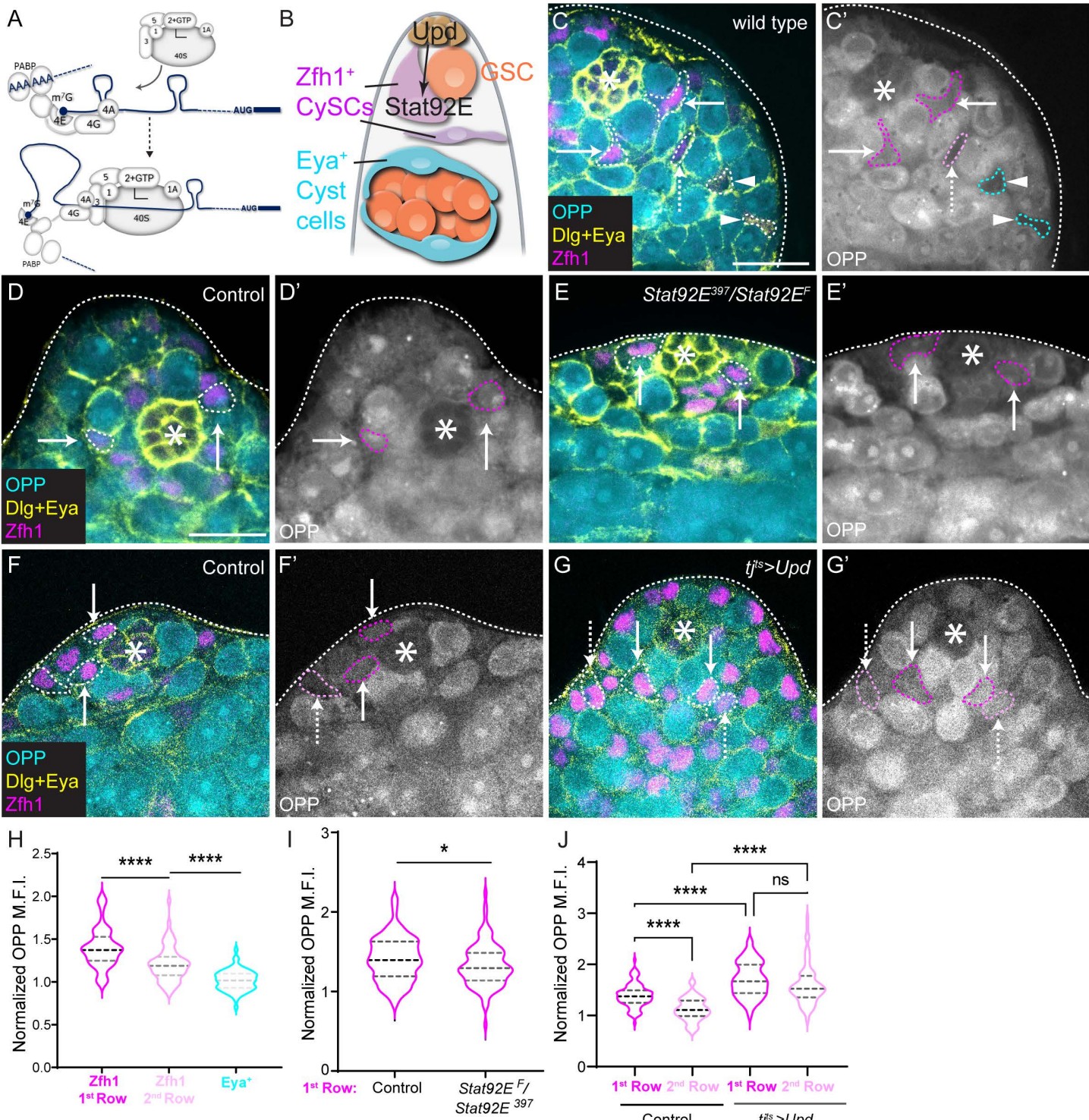

**Fig 1. JAK/STAT signaling promotes high global translation rates in CySCs.** (**A**) Schematic of translation initiation. The eIF4F complex, composed of eIF4A, 4E and 4G, binds the 5′ m⁷G cap and interacts with the 43S pre-initiation complex, which consists of the multi-subunit eIF3 complex, the eIF2-initiator Methionine tRNA-GFT complex, and eIF1, 1A and 5, together with the small 40S ribosome subunit. Interactions between eIF4F and eIF3 result in the ribosome being brought to the 5′ end of the mRNA and scanning for the start codon. (**B**) Schematic of a *Drosophila* testis. The hub (brown) produces the JAK/STAT ligand Unpaired (Upd) to activate Stat92E in surrounding stem cells. Two stem cell populations are supported by the hub, germline stem cells (GSC, orange), and somatic cyst stem cells (CySCs) (magenta). CySCs are found in two rows and express Zfh1; they differentiate into cyst cells that express Eya (cyan), which envelop differentiating germ cells (orange) and support their development. (**C–G**) OPP incorporation (cyan) was used to measure global translation rates in testes stained with antibodies against Dlg (yellow) to label the hub

and outline somatic cell membranes, Eya (yellow) to label fully differentiated cyst cells and Zfh1 (magenta) to label CySCs. Dashed lines outline CySCs adjacent to the hub (arrows), or in the 2nd row away from the hub (dashed arrows) and fully differentiated cells (arrowheads). Asterisks mark the hub. (C, C′) A wild-type testis showing highest OPP in CySCs adjacent to the hub and decreasing OPP levels with increasing distance from the hub. Scale bar: 15 μm. (D-E') A *control (+/Stat92E*397) testis (D, D′) and a testis from a temperature-sensitive *Stat92e* mutant (*Stat92EF/Stat92E*397) (E, E′) after 20h at the restrictive temperature. (F–G′) A control (*tj*ts> +) testis (F, F′) and a testis with Upd over-expression (*tj*ts> *upd*) (G, G′) after 20h at the restrictive temperature. (H) Quantification of OPP incorporation in wild-type testes comparing 1st row CySCs, 2nd row CySCs and Eya$^+$ cyst cells, normalized to levels in the hub ($N > 80$ cells from 10 testes, Student $t$ test, ****$P < 0.0001$). (I) Quantification of OPP incorporation in 1st row cyst cells (CySCs) normalized to levels in the hub. CySCs have significantly lower OPP levels in *Stat92e* mutant testes than in controls ($N > 100$ cells from $\geq 8$ testes, Student $t$ test, *$P < 0.05$). (J) Quantification of OPP incorporation in 1st row and 2nd row CySCs normalized to levels in the hub. Upd over-expression results in significantly higher OPP levels in both rows, and abolishes the difference between first and second row ($N > 80$ cells from $\geq 7$ testes, Šidák multiple comparisons test, ****$P < 0.0001$, ns $P = 0.0946$.) Underlying data for all graphs can be found in file S1 Data.

Kinase 2 (CK2) [28], although to date this regulation has only been demonstrated in culture and in response to stress, and it is unknown whether it occurs during homeostasis in tissues. Non-canonical translation has been shown to occur in stem cells or during differentiation [16,17], yet how translation modes can be regulated in response to niche signals to influence cell fate is still poorly understood.

To explore the relationship between niche signals and translational regulation, we use the *Drosophila* testis as a model, since the niche and its signaling are well-understood and can be manipulated with exquisite precision [29]. In the *Drosophila* testis, the stem cell niche, called the hub, is a cluster of 10–12 somatic cells that supports the self-renewal of two different stem cell populations: GSCs and somatic cyst stem cells (CySCs, Fig 1B) [29,30]. GSCs give rise to daughter cells known as gonialblasts, which divide with incomplete cytokinesis to form germline cysts that eventually produce spermatids, while CySCs give rise to post-mitotic cyst cells which envelop gonialblasts and are critical for the development of germ cells. CySC self-renewal is primarily regulated by the Janus kinase (JAK)/signal transducer and activator of transcription (STAT) pathway. Hub cells secrete the JAK/STAT ligand Unpaired (Upd), which leads to activation of the sole *Drosophila* STAT, Stat92E, in stem cells immediately adjacent to the hub [31,32]. Active Stat92E acts to promote transcription of several targets that encode important factors for CySC self-renewal, including *Zn-finger homeodomain 1* (*zfh1*), and Zfh1 is commonly used as a marker for CySCs [33–35]. As they differentiate, cyst cells express Eyes absent (Eya) (Fig 1B) [36]. The relative contributions of transcriptional and post-transcriptional regulation to these changes in gene expression downstream of JAK/STAT activation have not yet been determined.

Here, we examined global translation in somatic cells of the testis. We found that translation levels are higher in CySCs than differentiating cyst cells, and that JAK/STAT signaling maintains high translation rates. To understand how these rates were regulated, we knocked down translation initiation factors and found a specific role for the mRNA 5′ m$^7$G cap-binding eIF4F complex in maintaining self-renewal downstream of JAK/STAT, while most subunits of other initiation complexes were required for differentiation. Notably eIF3d1 was also required for self-renewal, and provides a potential mechanism for regulation of translation initiation since eIF3d is sensitive to phosphorylation [28]. Indeed, we show that a phospho-mimetic eIF3d1 can rescue loss of self-renewal upon Casein Kinase II (CkII) knockdown, while phospho-dead eIF3d1 cannot. Moreover, CK2 modulates the ability of eIF3d to interact with the eIF4F complex in human cells, suggesting that phosphorylation of eIF3d is a switch between different modes of translation initiation. Moreover, CkII over-expression is sufficient to functionally rescue CyCSs in the absence of JAK/STAT signaling, placing it as a key effector of self-renewal signaling. In sum, our data suggest that CySCs have a specific translational programme, requiring activity of the cap-binding eIF4F complex and that this activity is regulated by phosphorylation of eIF3d1 by CkII. The mechanism we uncover here suggests a model in which stem cells, loaded with messenger RNAs encoding both self-renewal and

differentiation factors, can make rapid fate choices by switching translational programme, ensuring specificity in expression downstream of signals from the niche.

## Results

### JAK/STAT activity maintains high levels of translation in CySCs

Studies in several stem cell lineages have revealed that global translation rates change during differentiation [16,17]. To monitor translation rates in the *Drosophila* testis, we incubated dissected testes with O-propargyl-puromycin (OPP) and visualized its incorporation in different cell types. Cells can be identified using molecular markers and by their position relative to the hub: CySCs contact the hub directly, and move further away as they differentiate [30]. Zfh1 labels CySCs and their immediate descendants, which include a population of cells that are licensed, but not committed to differentiate, while differentiated cells express Eya (Fig 1B) [2,33,36]. OPP levels were higher in Zfh1-positive CySCs adjacent to the hub (Fig 1C, arrows, quantified in Fig 1H), and appeared to decline as cyst cells progressed through differentiation such that Zfh1-positive cells away from the hub ("2nd row", Fig 1C dashed arrow, Fig 1H) had lower OPP levels than those adjacent to the hub (14% decrease, $P < 0.0001$), and Eya-positive cells had lower levels still (Fig 1C, arrowhead, Fig 1H, 16% decrease relative to 2nd row cells, $P < 0.0001$).

The difference in OPP incorporation between the first two rows of CySCs relative to the hub suggested that signals from the hub promote high translation rates in adjacent cells. However, an alternative possibility is that cells gradually differentiate as they leave the niche and that cells in the second row are at early stages of differentiation. To test directly whether translation rates were a consequence of differentiation state or of proximity to the niche, we inhibited differentiation in CySCs by knocking down the Retinoblastoma homologue Rbf using *tj$^{ts}$-Gal4*. We and others showed previously that Rbf knockdown results in expansion of the Zfh1-positive population away from the hub [37,38], and indeed, we observed Zfh1-positive cells away from the niche and an absence of Eya-positive cells (S1A Fig). OPP incorporation in ectopic Zfh1 cells away from the niche (S1A Fig, arrowheads) was lower than in cells adjacent to the hub (S1A Fig, arrows, quantified in S1B Fig, $P < 0.0001$). Thus, translation likely reflects proximity to the niche rather than differentiation status, consistent with several observations suggesting that the two rows of Zfh1-positive cells adjacent to the hub form a large pool of stem cells with frequent exchanges between the rows [2,39,40].

Given that JAK/STAT signaling is active in CySCs adjacent to the hub (Fig 1B, [33,35] and essential for their self-renewal, we wondered whether the sole *Drosophila* STAT, Stat92E, could be responsible for maintaining high levels of translation in these cells. Shifting adult flies carrying a temperature-sensitive *Stat92E* allelic combination, referred to as *Stat92E$^{ts}$*, to the restrictive temperature of 29 °C results in differentiation of both CySCs and GSCs; however, after 1 day of temperature shift, CySCs are still present, while GSCs begin to detach from the hub [2,41], allowing us to address the role of JAK/STAT in regulating translation in CySCs. As assessed by OPP incorporation, we observed that 20 h after temperature shift, global translation levels were decreased in Zfh1-positive CySCs immediately adjacent to the hub (Fig 1D, 1E and 1I, 6% decrease compared to control, $P < 0.05$), indicating that changes in translation occur following inactivation of Stat92E, but prior to cell fate changes. Conversely, since Zfh1-positive cells exhibit different rates of global translation depending on their position relative to the hub, which is the source of the JAK/STAT ligand Upd, we asked whether providing ectopic Upd would increase translation rates away from the hub. We used the cyst lineage driver *traffic jam (tj)-Gal4*, together with a temperature-sensitive Gal80 (together referred to as *tj$^{ts}$-Gal4*) to control the timing of expression and restrict it to adult stages.

Indeed, over-expression of Upd resulted in higher levels of OPP incorporation in Zfh1-positive cells, both those contacting the hub and cells in the second row from the hub (22% and 42% increase, respectively, $P < 0.0001$ for both relative to control, Fig 1F, 1G and 1J). OPP levels in 2$^{nd}$ row cells were indistinguishable from those in cells that contacted the hub (Fig 1J). Thus, increasing JAK/STAT activity elevates translation rates in Zfh1-positive CySCs.

Altogether, our results indicate that global translation rates decrease as CySCs move away from the hub, and that the main signaling pathway mediating self-renewal in CySCs, JAK/STAT, maintains high levels of translation in CySCs. Thus, we sought to determine whether increased translation was important for the self-renewal of CySCs.

## Distinct requirements for translation initiation factors in CySCs and differentiating cells

To test whether high translation rates were necessary to maintain CySC identity, we asked how CySCs were affected if translation was impaired. Initiation is the most highly regulated stage of mRNA translation and involves the assembly of several eIF complexes which recruit and activate ribosomes to scan the mRNA for the start codon (Fig 1A) [18]. Thus, by disrupting translation initiation, we expected to reduce global translation rates, and if high translation was required for CySC self-renewal, these manipulations should result in loss of CySCs. We used *tj*$^{ts}$-*Gal4* to knock down subunits of each eIF in somatic cells of the testis. Results of this screen are summarized in Fig 2A and shown in full in S1 Table.

Remarkably, we identified two opposite phenotypes, which clustered by complex. In the first, we observed either complete loss or severe reduction of CySC numbers, as assessed by Zfh1 expression. Knockdown of all subunits of the cap-binding eIF4F complex (eIF4A, eIF4E1, eIF4G1) as well as one subunit of eIF3, eIF3d1, resulted in significant decrease or complete absence of Zfh1-positive cells (Fig 2B–2E, quantified in Fig 2H and see below for eIF3d1). In contrast, all other knockdowns led to the presence of ectopic Zfh1-positive cells away from the hub (Fig 2F and 2G and S1 Table and see below).

These observations suggest that the eIF4F complex is necessary to maintain CySCs. To verify these findings and to test whether these phenotypes were cell autonomous, we generated positively-marked clones mutant for two independent alleles of *eIF4A*. Control clones were observed at 2 days post clone induction (dpci) as GFP-positive cells adjacent to the hub (S2A Fig, arrow). By 7 dpci, control clones were readily observed adjacent to the niche, indicating that the marked cells had self-renewed, and Eya-positive clonal cells were observed farther from the hub (S2B Fig, arrows and arrowhead, respectively), indicating that the clones gave rise to differentiating cyst cells. In contrast, *eIF4A* mutant CySCs were rarely recovered at 2 dpci, and never at 7 dpci (S2C–S2F Fig). This was not due to different induction rates, as marked cells (including both CySCs and differentiating cyst cells) were recovered at similar rates in mutant and control (S2E Fig). Instead, all *eIF4A* mutant cells observed were also Eya-positive at 7 dpci (S2D Fig, arrowhead), indicating that they had differentiated. To rule out the possibility that we did not observe *eIF4A* mutant CySCs because they were unable to synthesize GFP due to defective translation, we also generated negatively-marked clones. These clones showed similar results; control CySC clones lacking GFP were observed at the hub at 2 dpci and maintained at 7 dpci, while *eIF4A* mutant CySC clones lacking GFP were occasionally observed adjacent to the niche at 2 dpci and almost never at 7 dpci (S2G–S2J Fig). Additionally, we generated negatively-marked *eIF4E1* mutant clones, and observed a similar defect in mutant CySC clone recovery (S2K–S2N Fig), indicating that, like *eIF4A*, *eIF4E1* is required autonomously for CySC self-renewal. Two possibilities could explain the lack of marked mutant CySCs at 7 dpci, either CySCs lacking eIF4A or eIF4E1 died, or they differentiated into cyst cells. Since we observed Eya-positive mutant cyst cells, we favored the

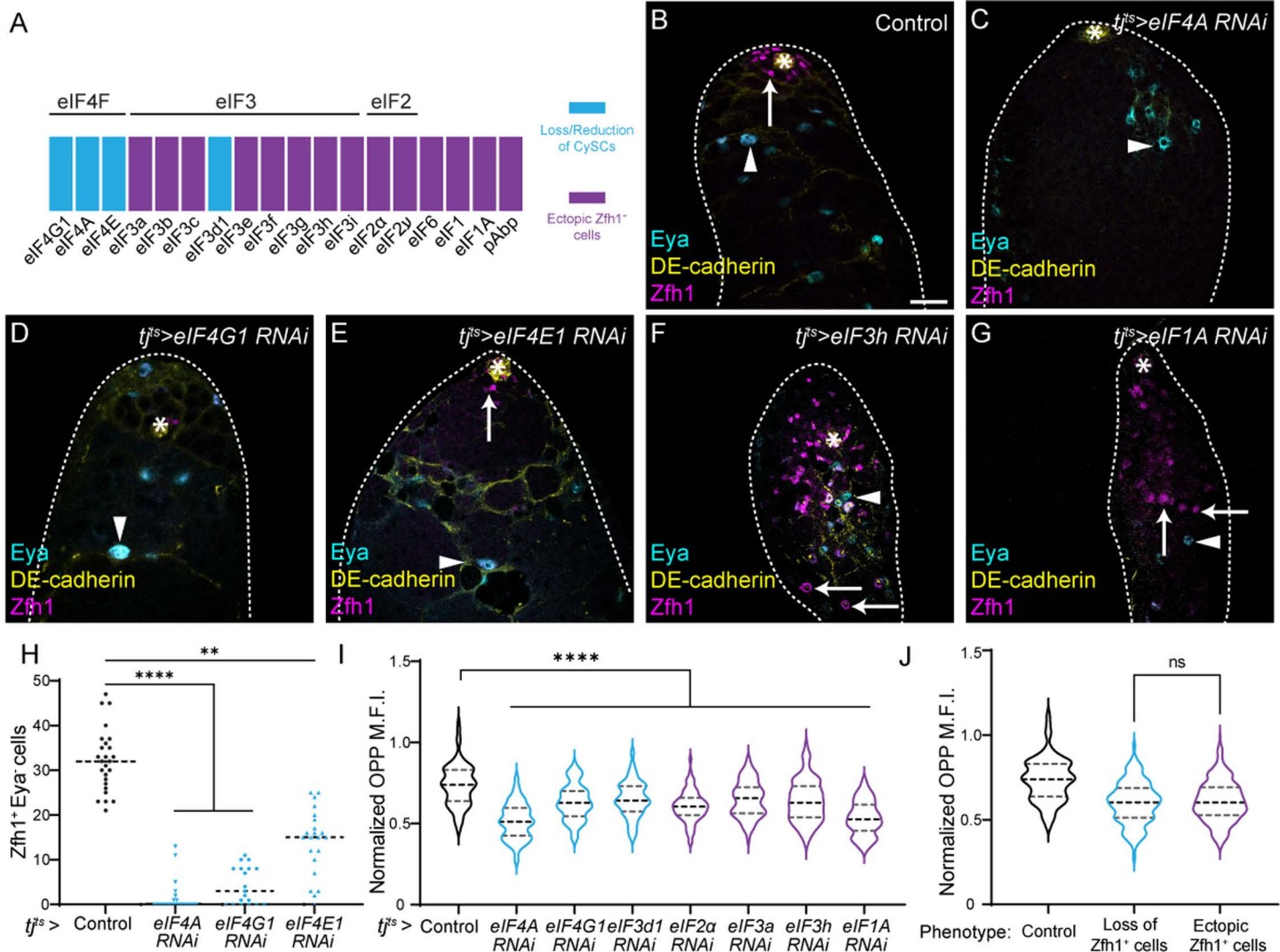

**Fig 2. Knockdown reveals different requirements for translation initiation factors in CySCs and differentiating cyst cells.** (**A**) Summary of the phenotypes observed upon knocking down eIFs. Knockdowns resulted either in complete loss or decrease of Zfh1-expressing cells (cyan), or ectopic presence of Zfh1-positive cells away from the hub (magenta). These phenotypes mostly clustered by complex, with the exception of eIF3d1. See S1 Table for full details. (**B–G**) Representative results from the initiation factor knockdown screen. A control (*tj*ts> +) testis (B), and testes in which the indicated eIF was knocked down (C-G). High DE-cadherin expression (yellow) labels the hub and lower levels label cell outlines. Zfh1 (magenta) labels CySCs and Eya (cyan) labels differentiated cyst cells. Asterisks indicate the hub. Arrows indicate CySCs and arrowheads indicate differentiated cells. Scale bar: 15 μm. (**H**) Quantification of the number of Zfh1+ Eya− cells upon knockdown of eIF4F complex subunits. ($N \geq 19$ testes, Kruskal-Wallis test, ****$P < 0.0001$, **$P < 0.01$.) (**I**) Quantification of OPP incorporation in CySCs adjacent to the hub in each of the indicated genotypes, normalized to levels in GSCs. Knockdowns resulting in CySC loss are shown in cyan, those resulting in ectopic CySCs are shown in magenta, all lead to decreased OPP incorporation ($N > 85$ cells from $\geq 8$ testes, Šidák multiple comparisons test, ****$P < 0.0001$.) (**J**) Grouped analysis of OPP levels in CySCs based on RNAi phenotype. No significant difference was observed between knockdowns resulting in CySC loss (cyan) or ectopic CySCs (magenta) ($N > 100$ cells from $\geq 8$ testes, Šidák multiple comparisons test, ns $P = 0.6764$.) Underlying data for all graphs can be found in file S1 Data.

second possibility. Nonetheless, we tested whether knockdown of eIF4G1 resulted in apoptosis of CySCs by staining testes with an antibody against the cleaved form of Death caspase-1 (Dcp-1). In controls, Dcp-1-positive dying cyst cells are readily observed (S3A Fig, arrows), but Dcp-1 staining is rarely, if ever, observed in CySCs adjacent to the hub [42]. We examined eIF4G1 knockdown testes after 7 days at the restrictive temperature, prior to complete CySC loss, and observed no increased Dcp-1 staining around the niche compared to control (S3B Fig). Since apoptotic cells may be cleared fast, we used the baculovirus caspase inhibitor

P35 to block apoptosis together with knockdown of eIF4G1 or eIF4A and assess whether this would prevent the loss of CySCs caused by eIF4F subunit loss. First, we validated that P35 could effectively prevent apoptosis, by counting testes containing Dcp-1 positive cells in controls and upon P35 expression, together with a suitable titration control for Gal4 activity (S3C–S3E Fig). Blocking cell death together with knockdown of eIF4G1 or eIF4A did not increase the number of CySCs compared to knockdown alone (S3F Fig), indicating that cell death is not the primary cause of the reduction in CySCs upon knockdown of eIF4F subunits. Finally, we asked whether cyst cells lacking eIF4A could differentiate normally. Cyst cells encapsulate germ cells and support their development; as cyst cells differentiate, they acquire a flattened morphology that extends and surrounds germ cell cysts (S4A Fig). We observed that *eIF4A* mutant cyst cells had a similar morphology, and germ cells encapsulated by these mutant cells appeared to develop normally (S4B and S4C Fig). Thus, we conclude that the cap-binding eIF4F complex is required specifically for self-renewal of CySCs, and that CySCs lacking eIF4F activity differentiate into functional cyst cells.

In contrast to the loss of CySCs observed upon eIF4F knockdown, knockdown of subunits of all the other translation initiation complexes resulted in the presence of ectopic Zfh1-positive cells away from the hub (Fig 2A and S1 Table, examples are shown in Fig 2F and 2G). These testes often also contained Eya-positive cells, indicating that some differentiation did occur. Nonetheless, the presence of ectopic Zfh1-positive cells away from the hub suggested that CySCs continued to self-renew when these translation initiation factors were knocked down. To confirm this, we assessed proliferation using the S-phase marker 5-ethynyl-2′-deoxyuridine (EdU), since CySCs are the only proliferating somatic cells in the testis [30,43,44]. In controls, somatic EdU incorporation is only observed in Zfh1-positive cells adjacent to the hub (S5A Fig, arrows). When eIF2 or eIF3 complex subunits, or other initiation factors such as eIF1, eIF1A and eIF6 were knocked down, we observed EdU incorporation in Zfh1-positive cells many cell diameters distant from the hub (S5B–S5D Fig, dashed arrows), indicating that the ectopic Zfh1-positive cells have stem-like properties. Thus, inhibiting translation initiation factors other than eIF4F results in ectopic CySC self-renewal away from the niche and defective, although not fully blocked, differentiation.

In sum, knocking down translation initiation factors produced two opposite phenotypes: either ectopic self-renewal when most initiation factors were knocked down, or a loss of CySC self-renewal when eIF4F was knocked down. There are two possibilities to explain how such different phenotypes could be obtained when translation is perturbed: either different fates are induced by different levels of global translation, or knocking down different initiation factors selectively affects the translation of specific mRNAs. Firstly, given that we observed higher levels of translation in CySCs than in cyst cells (Fig 1C and 1H), it is possible that these manipulations disrupt translation rates differently, such that eIF4F or eIF3d1 knockdown results in lower translation than other knockdowns, with a consequent effect on fate, where low translation rates lead to differentiation and high translation rates to self-renewal. To test this, we examined OPP incorporation in several conditions. All knockdowns we examined led to decreased translation relative to control, ranging from 12% for eIF3a knockdown to 30% decreased OPP incorporation for eIF4A knockdown compared to control (Fig 2I); importantly, knockdowns that led to ectopic Zfh1-positive cells and knockdowns that led to loss of Zfh1-positive cells decreased OPP incorporation to a similar extent, for instance eIF3a and eIF3d1 knockdowns reduced OPP by 12% and 13%, respectively, and eIF4A and eIF1A knockdowns reduced OPP by 30% and 28%, respectively. When taken together, there was no difference in the decrease in OPP incorporation across the two classes of phenotype (Fig 2J). Thus, we conclude that it is not the global rate of translation that determines cell fate.

## Expression of the differentiation marker Eya is post-transcriptionally regulated

The alternative possibility to explain how knockdown of different initiation factors can produce opposite phenotypes is that translation initiation factors can determine which mRNAs are translated; in other words, that there are different translational programmes in CySCs and in differentiating cyst cells, determined by the activity of different translation initiation factors. For this model, two assumptions must be true: (1) that mRNA transcripts encoding both CySC and cyst cell factors must be present in the same cells, but only translated when appropriate, and (2) that translation initiation factor activity is regulated by signals promoting cell fate decisions to enable a switch from one programme to another.

To test the first assumption, we examined transcript localization for *eya*. Eya protein is present in differentiating cyst cells and is absent from cells adjacent to the hub (Figs 1B and 3A) [36]. However, previous work showed that *eya* mRNA was detectable in sorted CySCs, while the *eya* enhancer drives expression in CySCs [33,37,45]. We therefore used in situ hybridization chain reaction (HCR) to examine the distribution of *eya* transcripts. We used *ImpL2-GFP* to identify CySCs [46,47]. As expected, we observed many punctae of *eya* signal in differentiating cyst cells (Fig 3B, arrowhead). Notably, *eya* transcripts were also detected in GFP-positive cells in the first and second rows from the hub (Fig 3B, arrows and inset), despite the fact that Eya protein is not detected in cells adjacent to the hub (Fig 3A, arrows). To confirm this result, we generated tumors consisting of only CySCs and GSCs by over-expressing Upd [33]. In these tumors, all somatic cells expressed ImpL2-GFP and Zfh1, while Eya protein was rarely, if ever detected (Fig 3C), consistent with previous observations [33,48]. In contrast, we consistently detected *eya* transcripts in somatic cells, marked by ImpL2-GFP (Fig 3D).

One possibility is that it is the levels of *eya* transcript that determine whether the protein is produced. Indeed, in control testes, *eya* mRNA levels in CySCs, although detectable, are much lower than in differentiating cyst cells (Fig 3B). To test this, we over-expressed *eya* using *tj-Gal4* and assessed mRNA and protein levels. Over-expression resulted in robust expression of *eya* mRNA in the entire cyst lineage, including CySCs contacting the hub (Fig 3E, arrows). Eya protein, by contrast, was only rarely detected in CySCs adjacent to the hub (Fig 3E, arrowhead shows an example of an Eya-expressing CySC contacting the hub) although we did detect precocious Eya expression in cells two rows from the hub. Thus, the mismatch between *eya* mRNA and Eya protein expression cannot simply be attributed to levels of expression.

Therefore, at least in the case of *eya*, transcripts encoding differentiation factors are expressed in stem cells, while their encoded protein is not, consistent with the existence of a selective translational programme in CySCs. We next set out to test the second assumption, that translation initiation is regulated downstream of niche signals.

## eIF4F functions downstream of JAK/STAT in CySC self-renewal

Thus far, our results indicate that there are different requirements for translation initiation factors in CySCs and differentiating cyst cells, and that transcripts encoding differentiation factors are present in stem cells, supporting a model in which regulation of initiation factors determines which transcripts are translated. Moreover, we showed that high global translation rates in CySCs depend on JAK/STAT activity (Fig 1). Since initiation is the rate-limiting step in translation, this suggests that the CySC self-renewal pathway JAK/STAT could act to regulate CySC-specific translation initiation. However, since both eIF4F and JAK/STAT loss-of-function result in loss of CySC self-renewal, another possibility is that eIF4F is required for the translation of JAK/STAT signal transduction components and that signaling is compromised

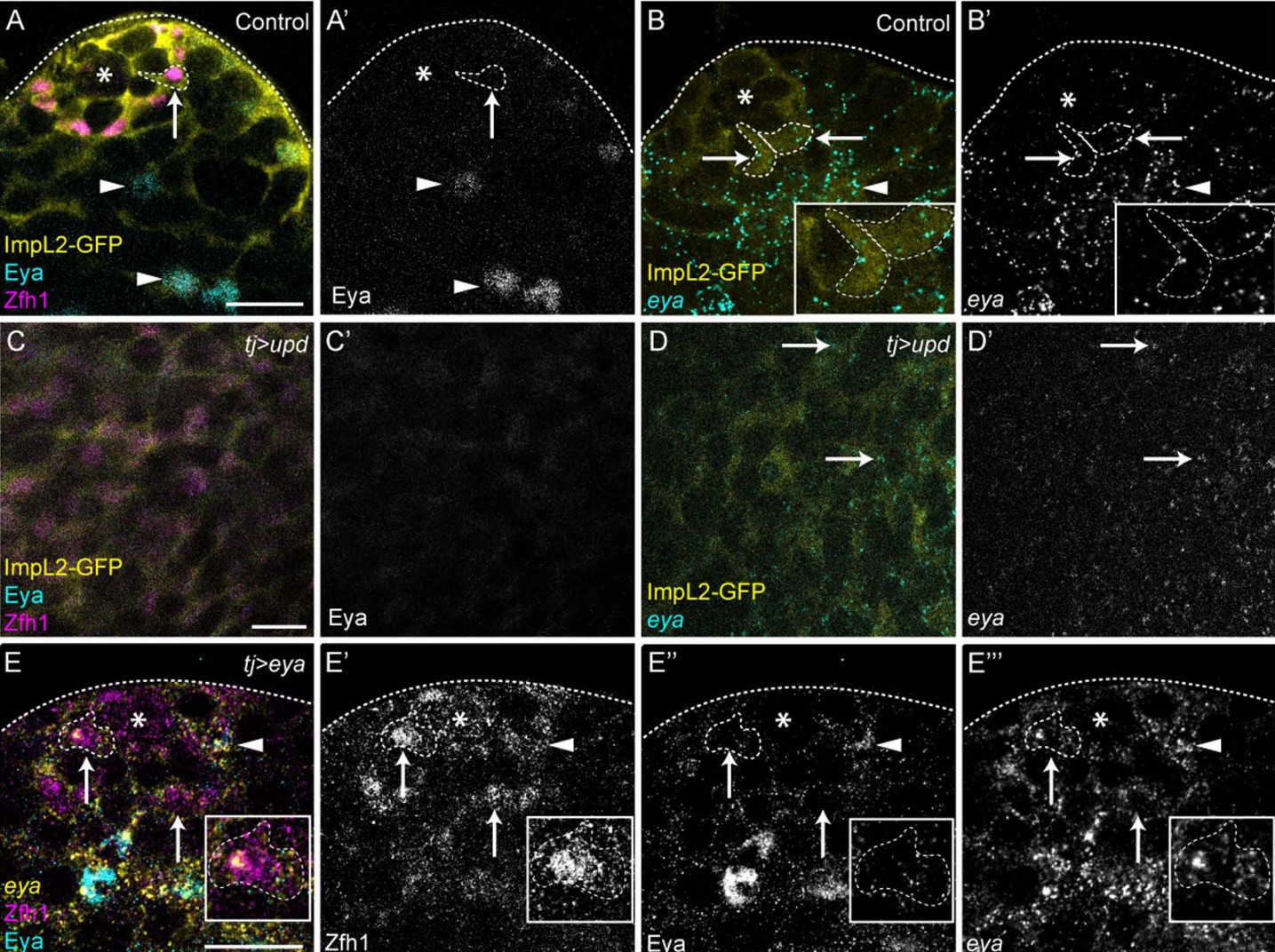

**Fig 3. Expression of *eya* is post-transcriptionally regulated in CySCs.** (A–D) Comparison of Eya protein (cyan A, C, single channel A′, C′) and *eya* mRNA (cyan B, D, single channel B′, D′) localization in control testes (A, B) or testes over-expressing Upd and containing stem cell tumors (C, D). ImpL2-GFP (yellow) was used to identify CySCs together with Zfh1 antibody in immunostainings (A, C). Transcripts were detected in CySCs in control and in Upd over-expressing testes where no Eya protein is visible. Asterisks indicate the hub. Arrows mark CySCs and arrowheads mark differentiated cells. Individual CySCs are outlined (A, B) and enlarged in the inset in B. Scale bar: 15 μm. N ≥ 15 testes. (**E**) Eya protein (cyan, single channel E″) and *eya* mRNA (yellow, single channel E‴) localization upon over-expression of *eya* in CySCs using *tj-gal4*. *eya* mRNA is present in CySCs around the hub, but Eya protein is not detected (arrows), although occasionally CySCs present both mRNA and protein (arrowhead). An individual CySC is outlined enlarged in the inset. Scale bar: 15 μm. *N* = 26 testes.

when eIF4F subunits are knocked down. To rule out this possibility, we examined levels of Stat92E, which is stabilized only in cells where signaling is active [49]. We generated clones marked by loss of GFP and measured Stat92E immunofluorescence in clones normalized to neighboring non-clonal cells (Fig 4A and 4B, quantified in Fig 4F). We did not observe any decrease in Stat92E levels in *eIF4A* null mutant CySCs compared to control CySCs. To further confirm that JAK/STAT signaling was not disrupted, we examined expression of the JAK/STAT pathway target, *Socs36E*, by in situ HCR. In control testes, *Socs36E* transcripts were detected in the hub and surrounding CySCs as expected (S6A Fig). Both the expression pattern and levels were unchanged when eIF4A was knocked down (S6B and S6C Fig), indicating

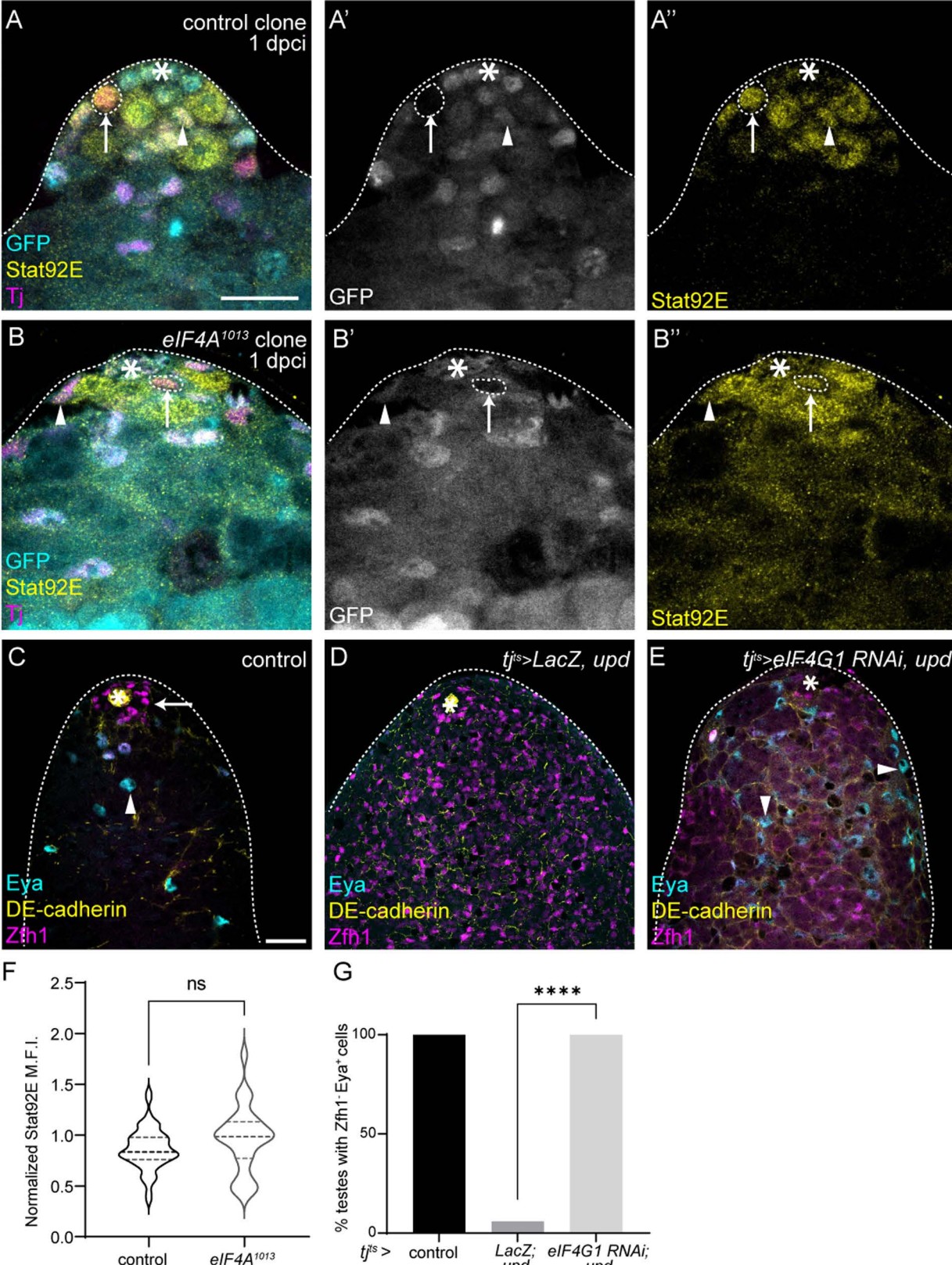

**Fig 4. eIF4F is epistatic to JAK/STAT in CySCs.** (A–B″) Testes with control clones (A–A″) and *eIF4A1013* clones (B–B″), 2 days post clone induction (dpci) labeled with an antibody against Stat92E (yellow, single channel A″,B″). Clones are marked by absence of GFP (cyan, single

channel A′,B′), and Tj (magenta) labels CySCs and early cyst cells. Dashed lines outline clonal CySCs (arrows) and arrowheads mark non-clonal CySCs. Mutant cells show normal Stat92E expression. Asterisks indicate the hub. Scale bar: 15 μm. (**C–E**) A control (*tj*ts> *LacZ*, +) testis (C), a testis in which Upd is over-expressed (*tj*ts> *LacZ, upd*) (D) and a testis with concomitant Upd over-expression and eIF4G1 knockdown in CySCs (*tj*ts > *eIF4G1 RNAi, upd*) (E). High levels of DE-cadherin (yellow) label the hub and low levels outline other cells, Zfh1 (magenta) labels CySCs and Eya (cyan) labels differentiated cyst cells. Upd over-expression results in stem cell tumors containing only Zfh1-positive cells and a lack of differentiated Eya-positive cells, while eIF4G1 knockdown rescues Eya-positive cells. Arrows indicate CySCs and arrowheads indicate differentiated cells. Asterisks indicate the hub. Scale bar: 15 μm. (**F**) Quantification of Stat92E levels in CySC clones (arrows in A–B″) of the indicated genotype, normalized to levels in neighboring non-clonal CySCs (arrowheads in A–B″) ($N \geq 28$ cells from ≥11 testes, Student $t$ test, ns $P = 0.0943$). (**G**) Percentage of testes with Eya-positive, Zfh1-negative differentiating cyst cells in the indicated genotypes ($N \geq 15$ testes, Chi-squared test, ****$P < 0.0001$). Underlying data for all graphs can be found in file S1 Data.

that JAK/STAT signaling is functional in cells lacking eIF4F subunits and cannot account for their self-renewal defect.

Next, we sought to establish whether translation initiation through eIF4F is required downstream of JAK/STAT signaling for CySC self-renewal. Over-expression of the JAK/STAT ligand Upd in testes using *tj*^*ts*^*-Gal4*, together with LacZ as a titration control, resulted in tumors containing many Zfh1-positive CySC-like cells, and an almost complete absence of Eya-positive differentiated cyst cells (Fig 4C and 4D). By contrast, knocking down eIF4G1 or eIF4A restored the presence of Eya-positive cells in all testes examined (Figs 4E, 4G, S6D and S6E), indicating that eIF4F is required for self-renewal even in the presence of ectopic niche signals. We note that ectopic Zfh1-positive stem cells were still present and intermingled with Eya-positive cells in these testes, indicating an incomplete suppression of the JAK/STAT gain-of-function phenotype, and suggesting that translation is not the only downstream effector of JAK/STAT signaling. Nonetheless, given the known role of JAK/STAT in regulating transcription, it is striking that modulating translation alone was sufficient to alter cell fate when JAK/STAT was hyperactivated.

## eIF3d1 is required for CySC self-renewal

Our results suggest that eIF4F activity is modulated such that it is specifically required for translation in CySCs, but not during differentiation, suggesting that there is a mechanism to selectively engage eIF4F in CySCs. One potential regulator of eIF4F activity is 4E-BP which binds eIF4E and prevents it from binding to the 5′ mRNA cap [50,51]. Target of Rapamycin (Tor) inactivates 4E-BP by phosphorylation, allowing eIF4E to interact with mRNAs [52,53]. However, we previously showed that 4E-BP is phosphorylated most highly in cells that leave the niche [2,46]. Therefore p4E-BP levels are high in cells that are likely to differentiate, which is inconsistent with the results described here in which eIF4F activity is specifically required for self-renewal. Thus, 4E-BP is unlikely to be the switch controlling eIF4F activity in CySCs.

While searching for an alternative mechanism that could be responsible, we noticed in our initiation factor screen above (Fig 2A and S1 Table) that, in addition to eIF4F, only one other initiation factor was required for CySC maintenance: eIF3d1, the *Drosophila* homologue of eIF3d. This was notable for two reasons: firstly, eIF3d is one of the subunits that mediates the interaction between eIF4F and eIF3 [54–56]. Secondly, in cell culture, phosphorylation of eIF3d has been shown to act as a molecular switch for eIF3d function, suggesting a mechanism by which eIF3d could alter translation in response to environmental changes [28]. Thus, to determine whether eIF3d1 could be regulated to control translation and maintain CySC self-renewal, we characterized the *eIF3d1* loss-of-function phenotype. Knockdown of eIF3d1 in adult testes resulted in loss of Zfh1-positive cells (Fig 5A and 5B), such that many testes were entirely devoid of CySCs (Fig 5C, $P < 0.0001$). We asked whether the loss of eIF3d1-deficient CySCs could be due to cell death. We did not detect Dcp-1 staining in CySCs when eIF3d1 was knocked down, and co-expressing the caspase inhibitor P35 did not rescue CySC numbers upon *eIF3d1* knockdown (S6F and S6G Fig), indicating that apoptosis is not the

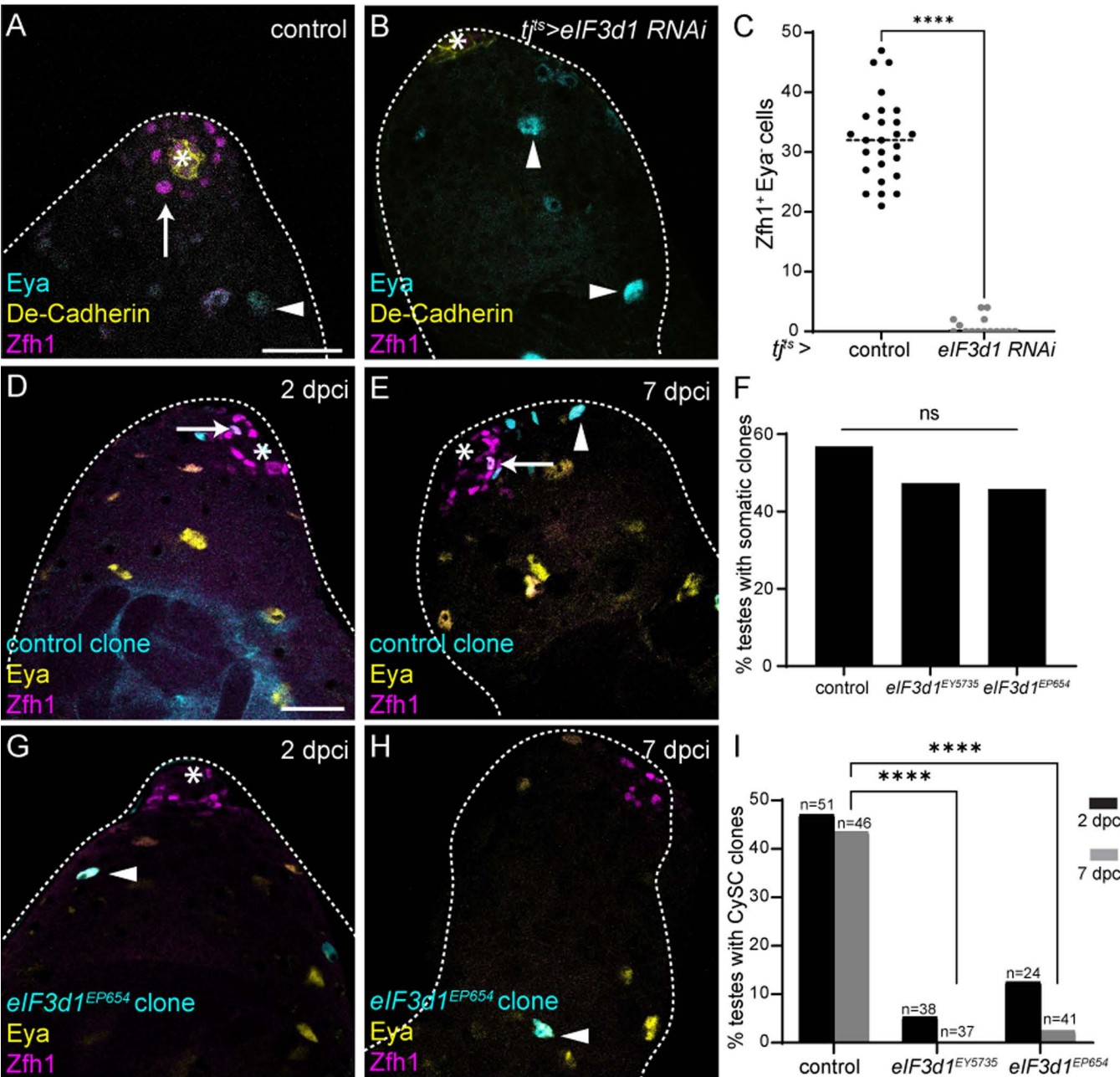

**Fig 5. eIF3d1 maintains CySC self-renewal.** (**A**, **B**) A control *(tj*ts> +) testis (A) and testis in which eIF3d1 was knocked down somatically (*tj*ts > *eIF3d1 RNAi*) (B). DE-cadherin (yellow) labels the hub. Zfh1 (magenta) labels CySCs and Eya (cyan) labels differentiated cyst cells. Arrows mark CySCs and arrowheads mark differentiated cells. Asterisks indicate the hub. Scale bar: 15 μm. (**C**) Number of Zfh1⁺ Eya⁻ cells in testes of the indicated genotype ($N \geq 16$ testes, Mann-Whitney test, ****$P < 0.0001$). (**D**, **E**) Testes with control clones at 2dpci (D) and 7dpci (E), marked by GFP expression (cyan). Zfh1 (magenta) labels CySCs and Eya (yellow) labels differentiated cells. Arrows mark clonal CySCs and arrowheads mark clonal differentiated cells. Asterisks indicate the hub. Scale bar: 15 μm. (**F**) Percentage of testes with clones in either CySCs or differentiated cyst cells at 2 dpci ($N \geq 24$ testes, Chi-squared test, ns $P = 0.5616$). (**G**, **H**) Testes with *eIF3d1*EP654 clones at 2dpci (G) and 7dpci (H) marked by GFP expression (cyan). Zfh1 (magenta) labels CySCs and Eya (yellow) labels differentiated cells. Arrowheads mark clonal differentiated cells; note that mutant CySCs are not observed. Asterisks indicate the hub. Scale bar: 15 μm. (**I**) Percentage of testes with marked CySC clones at 2 dpci and 7 dpci. In controls, clones are induced and maintained at 7 dpci, while for both alleles of *eIF3d1*, few mutant CySC clones are recovered ($N \geq 24$ testes, Chi-squared test, ****$P < 0.0001$). Underlying data for all graphs can be found in file S1 Data.

cause of CySC loss. To ensure this phenotype was autonomous to CySCs, we generated mutant clones homozygous for two independent alleles of *eIF3d1*. Control clones were observed adjacent to the hub at 2 dpci (Fig 5D) and maintained at 7 dpci, when they consisted of both Zfh1-positive cells and Eya-positive cells (Fig 5E, arrow and arrowhead, respectively). *eIF3d1* mutant clones were induced at similar rates to control (Fig 5F), but were rarely observed adjacent to the hub (Fig 5G). By 7 dpci, very few mutant CySC clones were recovered, and mutant cells were Eya-positive (Fig 5H, arrowhead, Fig 5I), indicating that they had differentiated. Altogether the clonal and knockdown data suggest that eIF3d1 is required autonomously in CySCs for self-renewal and that CySCs lacking eIF3d1 are lost from the niche due to differentiation, not cell death.

Next, we asked if, like eIF4F, eIF3d1 functioned downstream of JAK/STAT in promoting CySC self-renewal. We tested whether eIF3d1 knockdown affected JAK/STAT signal transduction by examining expression of *Socs36*E. We observed a similar expression pattern of *Socs36E* as in control testes, with high expression in the hub and surrounding CySCs, and when quantified, we could not detect any change in levels of expression upon eIF3d1 knockdown (S6C and S6H Fig). These results indicate that eIF3d1 is not required to maintain functional JAK/STAT signaling. We then asked whether eIF3d1 could act downstream of JAK/STAT, by knocking down eIF3d1 in testes in which Upd was over-expressed. Knockdown of eIF3d1 prevented the appearance of JAK/STAT-induced stem cell tumors, with 7/17 testes containing only Eya-positive cells, while 10/17 had both Eya-positive cells and some ectopic Zfh1-positive cells (S6E, S6I and S6J Fig). These data together suggest that eIF3d1 acts to promote CySC self-renewal downstream of JAK/STAT signaling and place eIF3d1 in a common genetic pathway with eIF4F.

## Phosphorylation of eIF3d1 by CkII maintains CySC self-renewal

Previous work has shown that phosphorylation of mammalian eIF3d can promote translation in two different modes: as part of the canonical eIF3 complex, which binds eIF4F and promotes eIF4E-dependent translation, or alternatively as a cap-binding protein that promotes translation when eIF4E is inactivated [25]. Phosphorylation of eIF3d has been shown to regulate its cap-binding activity: unphosphorylated eIF3d binds the cap while phosphorylated eIF3d does not [28]. In mammalian cells, CK2 (CkII in *Drosophila*) is the kinase responsible for phosphorylating eIF3d [28]. These phosphorylation sites are predicted to be conserved in *Drosophila* (S7A Fig).

To test the role of eIF3d phosphorylation, we generated UAS-eIF3d1 constructs in which the predicted CkII target serines were mutated to produce a phospho-mimetic form of eIF3d1 (UAS-eIF3d1^DD) or an unphosphorylatable form (UAS-eIF3d1^NN). Expression of wild-type or phospho-mimetic eIF3d1 had no effect on CySC numbers (S7B Fig); however, expression of phospho-dead eIF3d1 resulted in a 15% decrease in the number of CySCs ($P < 0.05$), suggesting that this construct acts as a dominant-negative. To confirm this, we examined OPP incorporation in CySCs, and observed that expression of phospho-dead eIF3d1 reduced OPP incorporation, while the phospho-mimetic had no significant effect on translation rates (S7C Fig). eIF3d can bind the mRNA 5′ cap directly to drive translation of specific transcripts [25,57]. To determine whether eIF3d1 cap binding was required in CySCs, we expressed a mutant form of eIF3d1, eIF3d1^helix11, which lacks the ability to bind the cap and which can act as a dominant-negative construct when eIF3d1 cap-binding function is required [25,58]. However, expression of this construct did not affect CySC numbers (S7D Fig). These results suggest that phosphorylation of eIF3d1 is required to maintain translation in CySCs and promote self-renewal.

We therefore tested whether the *Drosophila* CK2 homologue, CkII, composed of α and β subunits, was required for eIF3d1 function in CySCs. Knockdown of either *CkIIα* or *CkIIβ* using two independent RNAi constructs with *tjᵗˢ-Gal4* led to a dramatic loss of Zfh1-positive CySCs, with many testes entirely devoid of CySCs although Eya-positive cyst cells were observed (Fig 6A–6D). Thus, CkII is required for CySC self-renewal, similarly to eIF3d1. To determine whether CkII affected translation in CySCs, we measured OPP incorporation in CySCs after 1 day of *CkIIβ* knockdown, when CySCs were still present at the hub. The rate of global translation was significantly reduced in CySCs in which *CkIIβ* was knocked down compared to controls (Fig 6E, 14% reduction compared to control, $P < 0.0001$), suggesting that decreases in translation occur upon CkII knockdown prior to loss of CySCs.

Next, we asked whether phosphorylation of eIF3d1 was required downstream of CkII for CySC self-renewal. We reasoned that if the role of CkII was to phosphorylate eIF3d1, then providing an exogenous source of phosphorylated eIF3d1 should rescue the loss of CySCs caused by *CkII* knockdown. In testes in which *CkIIα* was knocked down alone, along with a suitable titration control, CySC numbers were reduced compared to control ($P < 0.0001$, Fig 6F). Expression of phospho-dead eIF3d1 had no effect, with CySC numbers similar to the knockdown control (UAS-eIF3d1^NN, $P = 0.29$, Fig 6F). By contrast, expression of phospho-mimetic eIF3d1 led to a significant rescue of CySCs (UAS-eIF3d1^DD, $P < 0.01$, Fig 6F). Unexpectedly, expression of wild-type eIF3d1 also led to a significant rescue (UAS-eIF3d1^WT, $P < 0.0001$, Fig 6F), suggesting that other kinases may regulate eIF3d1 phosphorylation non-redundantly with CkII, but that eIF3d1 levels are limiting in this context.

The fact that eIF4F is required for CySC self-renewal suggests that canonical cap-dependent translation occurs in CySCs. Loss-of-function of CkII or of eIF3d1 phenocopy eIF4F loss suggesting that it is not the cap-binding activity of eIF3d1 that is required in CySCs. These observations suggest a new model for eIF3d1 regulation by CkII, where phosphorylation of eIF3d1 is necessary to maintain the interaction between eIF4F and eIF3d1, in addition to the previously described role for this modification in modulating the cap-binding ability of eIF3d1 [28]. This would result in eIF3d1 acting as a switch to mediate recruitment of the 43S ribosome to mRNAs bound to eIF4F and would be consistent with loss-of-function of CkII, eIF3d1 and eIF4F subunits having similar phenotypes.

To test this hypothesis, we asked whether CK2 activity influenced the ability of eIF3d to interact with eIF4F using human cells in culture. We immunoprecipitated FLAG-tagged eIF3d and found, as expected, that it could co-immunoprecipitate eIF4A and eIF4G1 (Fig 6G), consistent with structural models and biochemical data placing eIF3d at the interface between the eIF3 and eIF4F complexes [54–56]. Next we treated cells with the CK2 inhibitor CX-5011, and repeated the pull-downs. We confirmed that CX-5011 inhibited CK2 by blotting total cell lysates with a pan-phospho-CK2 substrate antibody (S7E Fig). In these cells, CK2 inhibition resulted in a significant decrease in the ability of FLAG-tagged eIF3d to co-precipitate eIF4A and eIF4G1, (37% and 24% decrease IP relative to input, $P < 0.005$ and $P < 0.037$, respectively, Fig 6G and 6I). Notably, the interaction between eIF3d and other subunits of the eIF3 complex, eIF3b, eIF3a and eIF3l, was not decreased by CK2 inhibition (Fig 6G and 6I). Thus, CK2 is required for maximal association of eIF3d with eIF4F, but eIF3d association with the eIF3 complex appears independent of regulation by CK2.

To determine that phosphorylation of eIF3d itself by CK2 modulated the interaction between eIF3d and eIF4F, we repeated immunoprecipitations when over-expressing the phospho-mimetic form of eIF3d (Fig 6H). Like wild-type eIF3d, eIF3d^DD could co-precipitate eIF4A and eIF4G1, as well as eIF3 subunits; however, this interaction was no longer inhibited by addition of the CK2 inhibitor CX-5011 (Fig 6H and 6J), indicating that eIF3d^DD is insensitive to CK2 activity.

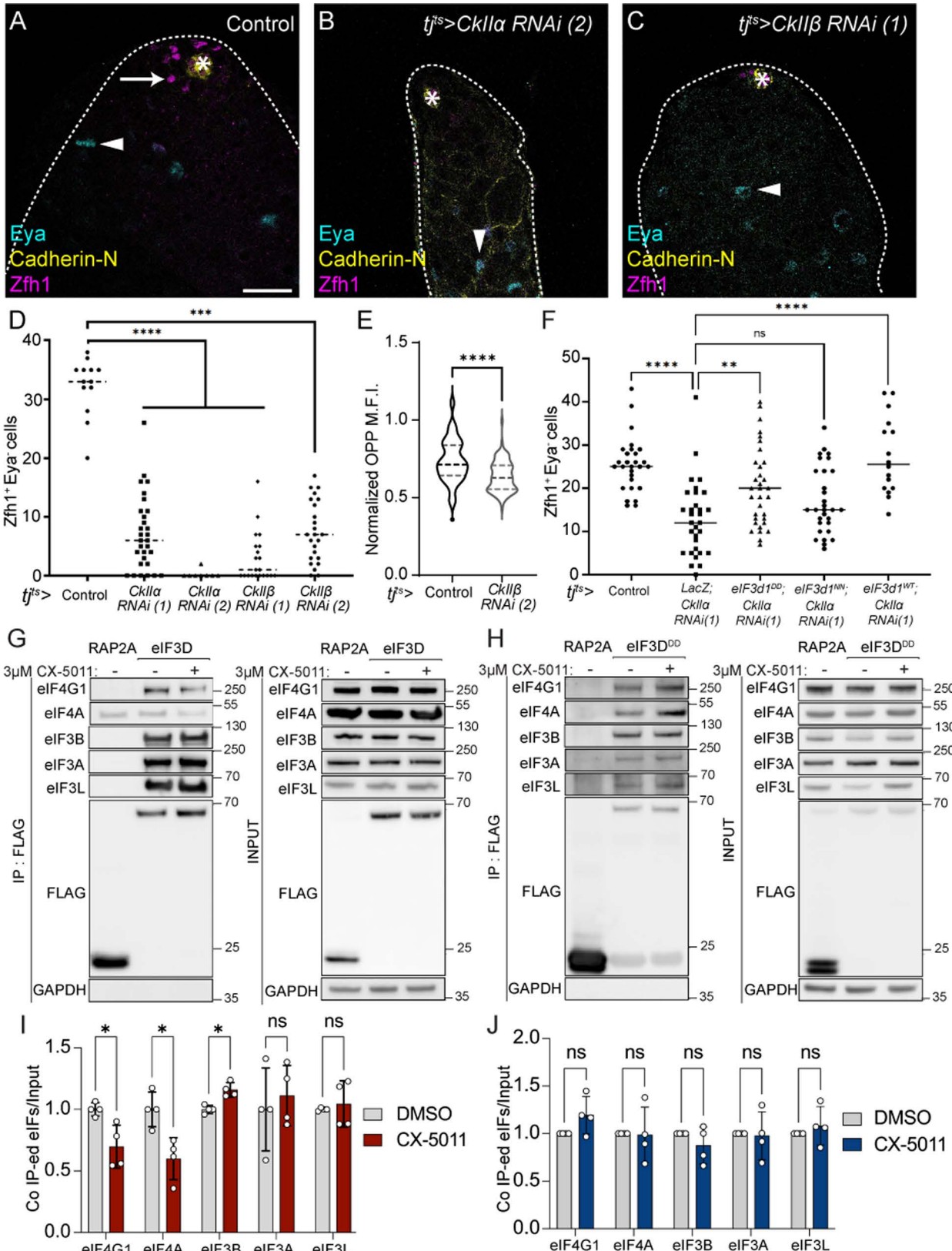

**Fig 6. CkII promotes the interaction between eIF3d1 and eIF4F to maintain CySC self-renewal.** (**A–C**) A control (*tj*ts> *LacZ, +*) testis (A), and testes in which CkIIα (B) or CkIIβ was knocked down (C). Cadherin-N (yellow) labels the hub. Zfh1 (magenta) labels CySCs and Eya

(cyan) labels differentiated cyst cells. Arrows mark CySCs and arrowheads mark differentiated cells. Asterisks indicate the hub. Scale bar: 15 μm. (**D**) Number of Zfh1⁺ Eya⁻ CySCs in testes of the indicated genotypes. Two independent RNAi lines were used to knock down both CkIIα and CkIIβ, leading to significant loss of CySCs ($N \geq 7$ testes, Kruskal–Wallis test, ****$P < 0.0001$, ***$P < 0.001$). (**E**) Quantification of OPP incorporation in CySCs adjacent to the hub in control or CkIIβ knockdown, normalized to levels in GSCs ($N > 70$ cells from $\geq 8$ testes, Student $t$ test, ****$P < 0.0001$). (**F**) Number of Zfh1⁺ Eya⁻ CySCs in testes in which CkIIα was knocked down, together with over-expression of phospho-mimetic (eIF3d1$^{DD}$), phospho-dead (eIF3d1$^{NN}$) or wild-type eIF3d1 (eIF3d1$^{WT}$). Phospho-mimetic and wild-type eIF3d1 expression significantly rescued CySC numbers while phospho-dead eIF3d1 did not ($N \geq 16$ testes, Kruskal–Wallis test, ****$P < 0.0001$, **$P < 0.01$, ns $P = 0.2892$). (G) Western blot from lysates of cells transfected with a control construct (RAP2A-FLAG) or with wild-type eIF3D-FLAG, blotted with antibodies against eIF4G1, eIF4A, eIF3B, eIF3A, eIF3L and FLAG, either in total lysate (input, right) or after immunoprecipitation with an anti-FLAG antibody (IP, left). Cells in the right lane were incubated with the CK2 inhibitor CX-5011 prior to lysis resulting in reduced co-immunoprecipitation of eIF4G1 and eIF4A with eIF3d-FLAG compared to untreated cells (middle lane). (H) Western blot from lysates of cells transfected with a control construct (RAP2A-FLAG) or with phospho-mimetic eIF3D$^{DD}$-FLAG, blotted with antibodies against eIF4G1, eIF4A, eIF3B, eIF3A, eIF3L and FLAG, either in total lysate (input, right) or after immunoprecipitation with an anti-FLAG antibody (IP, left). Cells in the right lane were incubated with the CK2 inhibitor CX-5011; no change in co-immunoprecipitation of eIF4G1 and eIF4A with eIF3D$^{DD}$-FLAG was visible compared to untreated cells (middle lane). (I) Quantification of immunoprecipitation blots showing reduced co-immunoprecipitation of eIF4G1 and eIF4A with eIF3d-FLAG, but not of eIF3 subunits, upon incubation with CX-5011. ($N = 4$ replicates, Student $t$ test, *$P < 0.016$). (J) Quantification of immunoprecipitation blots showing that incubation with CX-5011 does not affect co-immunoprecipitation of eIF4G1, eIF4A or eIF3 subunits with eIF3d$^{DD}$-FLAG. ($N = 4$ replicates, Student $t$ test, ns $P = 0.1$ (4G1), ns $P > 0.9$ (4A)). Underlying data for all graphs can be found in file S1 Data, raw western blot images can be found in file S1 Raw Images.

Altogether, our data suggest a new model for translational regulation by eIF3d1 in CySCs: CkII phosphorylates eIF3d1 to promote its interaction with the eIF4F complex.

## CkII is epistatic to JAK/STAT in CySCs

Since the JAK/STAT pathway is the main regulator of self-renewal and since we showed above that its activity maintains high translation in CySCs, we asked whether JAK/STAT could regulate translation and self-renewal through CkII. First, we tested whether CkII was required genetically downstream of JAK/STAT to maintain CySC self-renewal. We over-expressed the JAK/STAT pathway ligand Upd using *tj^ts*-Gal4, giving rise to tumors composed of CySCs expressing Zfh1 and devoid of Eya-expressing cyst cells (Fig 7A). When *CkIIα* was knocked down in these testes, only Eya-positive cells were detected, and no Zfh1-expressing CySCs were present (Fig 7B and 7C), although in rare cases double positive differentiating cyst cells were observed (Fig 7C). Thus, CkII is required downstream of JAK/STAT for CySC self-renewal. We ruled out the possibility that this requirement was because CkII affected the transduction of JAK/STAT signaling by monitoring levels of Stat92E and saw no difference between control and CkII knockdown CySCs (S8A–S8C Fig), as well as no change in levels of the JAK/STAT target *Socs36E* (S8D and S8E Fig).

Next, we tested whether expressing the catalytic subunit of CkII, CkIIα, was sufficient to maintain CySC function in the absence of JAK/STAT signaling. We used the *Stat92E^ts* mutant and observed after 10 days at the restrictive temperature that there were few CySCs around the hub, albeit in this genetic background, CySCs were not completely lost (Fig 7D). Expression of CkIIα with *tj^ts*-Gal4 led to an increase in CySC numbers (Fig 7E and 7F, $P < 0.01$), suggesting that CkII can sustain CySC self-renewal in the absence of JAK/STAT signaling. Strikingly, we noticed that testes from *Stat92E^ts* mutants in which CkIIα was over-expressed appeared to contain many more cells. In *Stat92E^ts* mutants, GSCs are lost as well as CySCs, but previous work showed that restoring Stat92E function only in CySCs was sufficient to maintain both stem cell populations [41]. Therefore, we examined germ cell presence and morphology using an antibody against Vasa in these testes. In agreement with previous findings, *Stat92E^ts* mutants shifted to the restrictive temperature for 10 days lacked GSCs, identified as individual round cells in contact with the hub, and were often devoid of spermatogonia entirely (Fig 7G–7I) [41]. In contrast, testes in which CkIIα was expressed often contained GSCs and appeared full of spermatogonia and spermatocytes (Fig 7H and 7I). Thus, expression of CkIIα

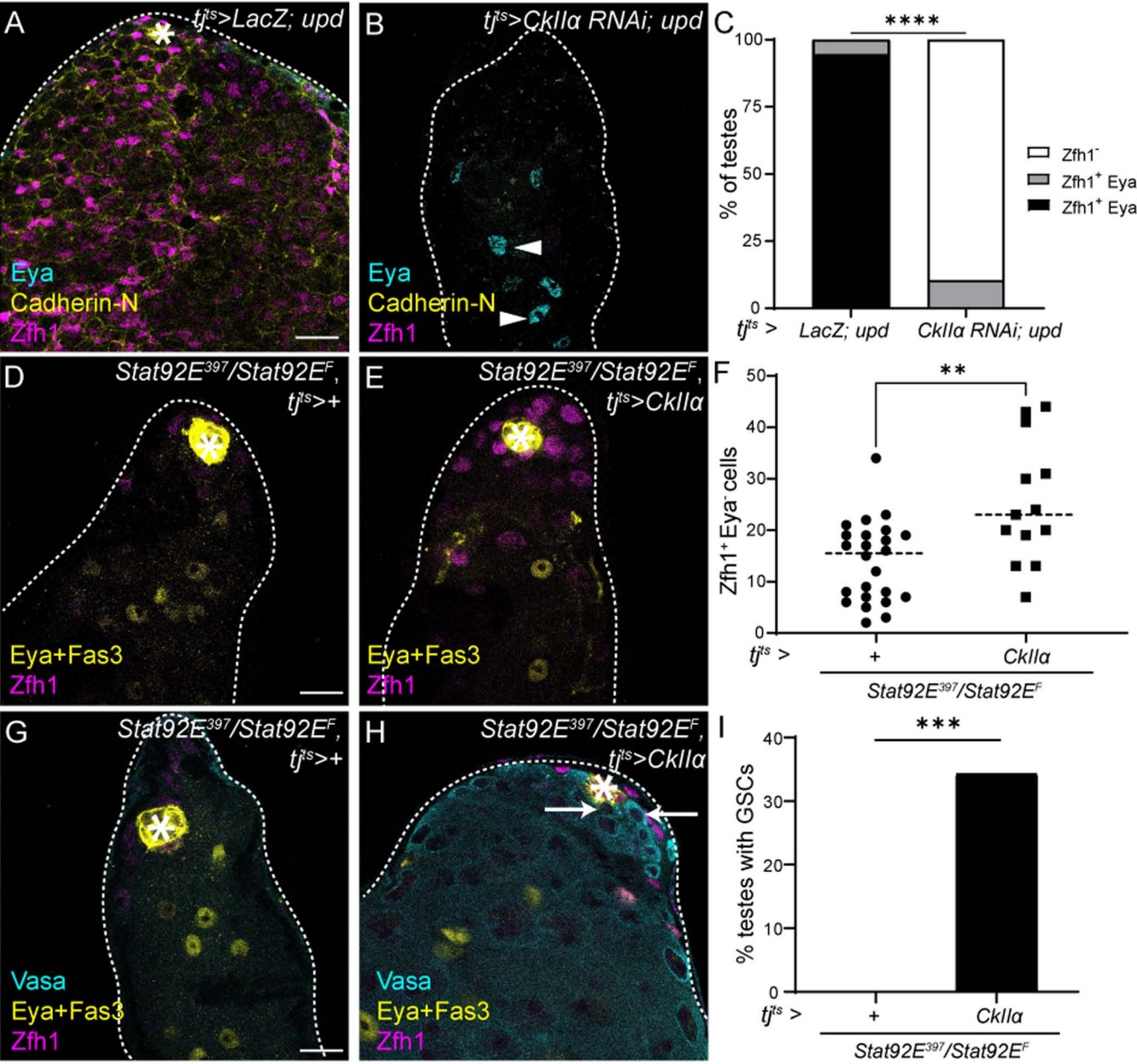

**Fig 7. CkII mediates both autonomous and non-autonomous functions of JAK/STAT signaling in CySCs.** (**A**, **B**) A testis in which JAK/STAT is hyperactivated (*tj*ts> *LacZ, upd*) contains only CySCs and is devoid of differentiating cyst cells (A) while a testis in which CkIIα is knocked down in CySCs concomitantly with JAK/STAT hyperactivation (*tj*ts> *CkIIα RNAi; upd*) contains only Eya-positive differentiated cyst cells (B). Cadherin-N (yellow) labels the hub. Zfh1 (magenta) labels CySCs and Eya (cyan) labels differentiated cyst cells. Arrows mark CySCs and arrowheads mark differentiated cells. Asterisks indicate the hub. Scale bar: 15 μm. (**C**) Percentage of testes containing only CySCs (Zfh1⁺ Eya⁻, black), both CySCs and cyst cells (Zfh1⁺ Eya⁺, gray) and no CySCs (Zfh1⁻, white) in the indicated genotypes ($N \geq 18$ testes, Chi-squared test, ****$P < 0.0001$). (**D**, **E**) A testis from a temperature-sensitive *Stat92e* mutant (*tj*ts>+; *Stat92E*F/*Stat92E*397) (D) and a testis from a temperature-sensitive *Stat92e* mutant expressing CkIIα (*tj*ts> *CkIIα, Stat92E*F/*Stat92E*397) (E) after 10 days at the restrictive temperature. CkIIα expression results in increased numbers of CySCs around the hub. Fas3 (yellow) labels the hub, Zfh1 (magenta) labels CySCs and Eya (yellow) labels differentiated cyst cells. Asterisks indicate the hub. Scale bar: 15 μm. (**F**) Number of Zfh1⁺ Eya⁻ CySCs in testes of the indicated genotype ($N \geq 13$ testes, Mann–Whitney test, ** $P < 0.01$). (**G,H**) Germ cells (Vasa, cyan) are absent in a testis from a temperature-sensitive *Stat92e* mutant (*tj*ts>+; *Stat92E*F/*Stat92E*397) (G) but are clearly visible in a testis from a temperature-sensitive *Stat92e* mutant expressing CkIIα (*tj*ts> *CkIIα, Stat92E*F/*Stat92E*397) (H) after 10 days at the restrictive temperature. Fas3 (yellow) labels the hub. Zfh1 (magenta) labels CySCs and Eya (yellow) labels differentiated cyst cells. Arrows indicate GSCs, identified as individual round Vasa-positive cells adjacent to the hub. Scale bar: 15 μm. (**I**) Percentage of testes containing any GSCs in temperature-sensitive *Stat92e* mutants alone or expressing CkIIα (N ≥ 30 testes, Chi-squared test, ***$P < 0.001$). Underlying data for all graphs can be found in file S1 Data.

in CySCs was sufficient to compensate for both autonomous functions of JAK/STAT in CySC self-renewal and non-autonomous functions in maintaining GSCs.

We sought to establish how CkII activity was regulated. The simplest explanation, given that CkII is genetically downstream of the Stat92E transcription factor, was that JAK/STAT signaling could regulate expression of CkII subunits. We examined expression of *CkIIα* using in situ HCR in control testes (Fig 8A). *CkIIα* was present in both somatic and germ cells, but using ImpL2-GFP to outline CySCs and cyst cells revealed that its expression decreased in cyst cells relative to CySCs (Fig 8A and 8B, $P < 0.001$). Next, we asked if this high expression in CySCs depended on JAK/STAT signaling by examining *CkIIα* levels in testes of *Stat92E^ts^* anim*a*ls shifted to the restrictive temperature for 1 day. Indeed, CySCs in *Stat92E^ts^* testes had 27% lower expression than control (Fig 8C–8E, $P < 0.001$), indicating that JAK/STAT signaling promotes expression of at least one CkII subunit. Altogether, these results indicate that JAK/STAT promotes CySC self-renewal through the regulation of CkII.

## Discussion

Despite extensive evidence that translation is regulated in stem cells to influence fate decisions, there is little understanding to date of how this regulation is achieved in response to signals from the stem cell niche. Here, using the somatic CySCs of the *Drosophila* testis as a model, we uncover a mechanism to regulate translation downstream of niche signals, and we propose that such post-transcriptional control achieves two goals: firstly, it allows stem cells to selectively translate (or not translate) mRNAs important for determining cell identity, ensuring that transcriptional noise does not result in aberrant protein expression; secondly, by enabling cells to switch between translational programmes and by having pre-existing expression of mRNAs encoding differentiation factors, this regulation allows plasticity and rapid adaptation to the environment. In particular, we show that global translation levels in CySCs depend on the self-renewal pathway JAK/STAT, and that CySCs and differentiated cyst cells have different requirements for translation initiation factors. The eIF4F complex is specifically required in CySCs for self-renewal but not differentiation, and acts genetically downstream of JAK/STAT signaling. We identify eIF3d1, the *Drosophila* homologue of eIF3d as a potential regulator of translation modes. We show that eIF3d1 is required for CySC self-renewal, like eIF4F, and that its phosphorylation by CkII is important for this function. Finally, we show that CkII acts downstream of JAK/STAT in CySCs, and that over-expressing CkII is sufficient to rescue both autonomous and non-autonomous roles of JAK/STAT signaling in CySCs. Altogether, we propose a model in which CkII and eIF3d1 act to switch between different modes of translation (Fig 9): when JAK/STAT is active in CySCs, eIF3d1 is phosphorylated by CkII and promotes eIF4F-dependent translation. We hypothesize that eIF4F drives selective translation of mRNAs encoding self-renewal factors, and does not translate mRNAs encoding differentiation factors, such as *eya*. As CySCs leave the niche and differentiate, JAK/STAT is inactive, leading to decreased CkII activity and consequently decreased eIF3d1 phosphorylation. Under these circumstances, eIF4F activity is reduced, enabling the translation of mRNAs encoding differentiation factors.

### Different modes of translation initiation during stem cell differentiation

Our work indicates that stem cells and differentiated cells have different requirements for initiation factors. Cells lacking eIF4F appear to differentiate into functional cyst cells, suggesting it is dispensable and that translation initiation in cyst cells does not require canonical cap-binding activity. This is consistent with the global decrease in translation rates during cyst cell differentiation, as cap-independent translation mechanisms are thought to be less efficient than canonical initiation by recruitment of the ribosome through eIF4F. Intriguingly, although

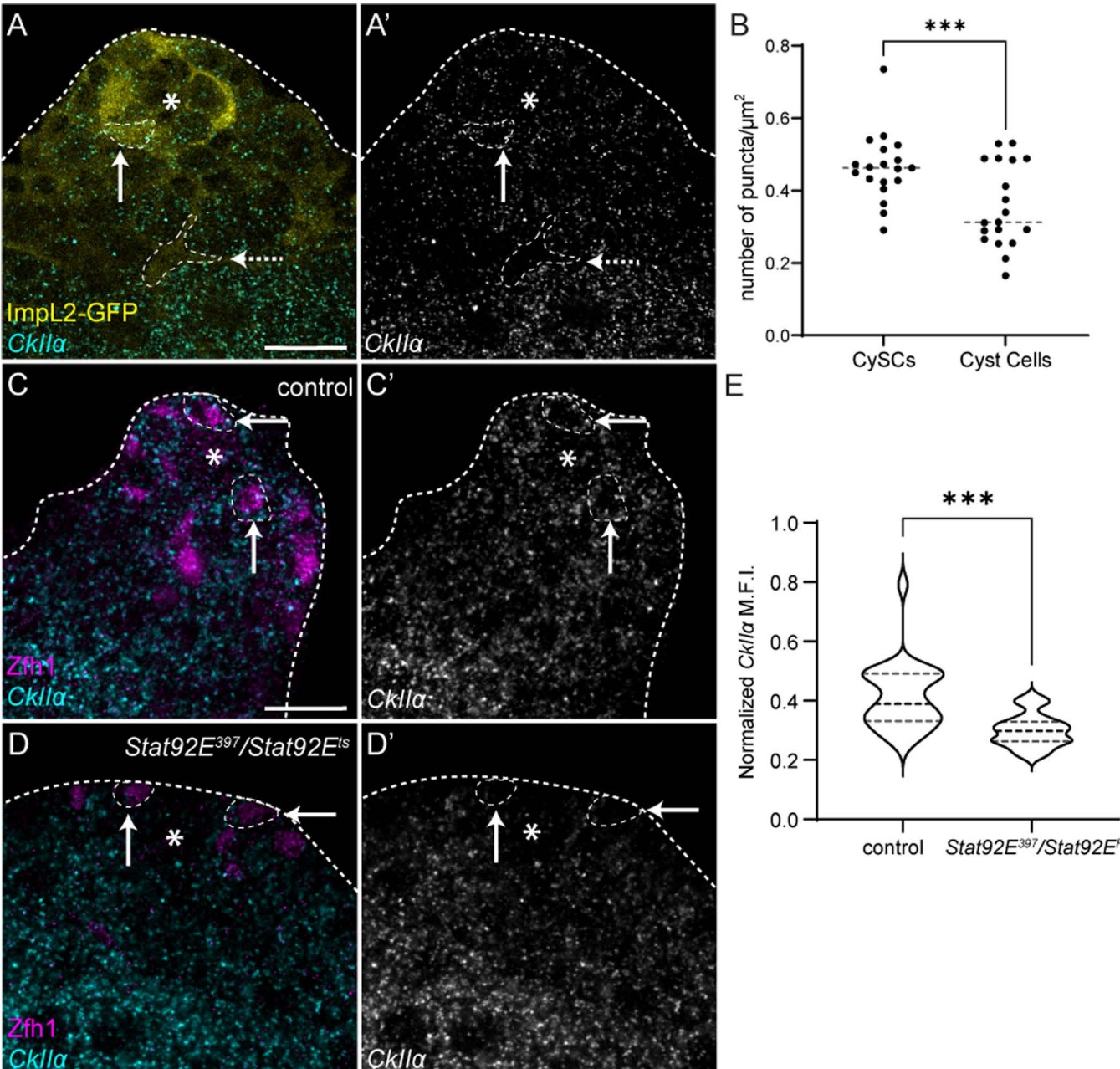

**Fig 8. *CkII α* expression in CySCs depends on JAK/STAT pathway activity.** (**A**) *CkIIα* expression detected by in situ hybridization chain reaction (cyan, single channel **A′**) in a testis from a control animal. ImpL2-GFP labels both CySCs and differentiated cyst cells. Solid arrow points to a CySC and dotted arrow points to a differentiated cyst cell. Asterisks indicate the hub. Scale bar: 15 μm. (**B**) Quantification of number of *CkIIα* puncta in CySCs and cyst cells per μm² in the confocal slice in which the largest hub section was visible. Levels in CySCs were measured by outlining ImpL2-positive regions adjacent to the hub and levels in cyst cells by outlining ImpL2-positive regions away from the hub before spermatocyte stages. ($N = 19$ testes, paired Student *t* test, ***$P < 0.001$) (**C, D**) *CkIIα* expression detected by in situ hybridization chain reaction (cyan, single channel **C′, D′**) in a control testis (C) and a testis from a *Stat92E*ts mutant animal after 1 day at the restrictive temperature (D). Zfh1 (magenta) labels CySCs (arrows). Asterisks indicate the hub. Scale bar: 15 μm. (**E**) Quantification of *CkIIα* probe fluorescence intensity in all CySCs around the hub in control and *Stat92E*ts testes, normalized to levels in spermatocytes ($N \geq 21$ testes, Student *t* test, ***$P < 0.001$). Underlying data for all graphs can be found in file S1 Data.

global translation rates change differently in different stem cell populations (for instance hematopoietic stem cells have lower translation rates than their differentiating offspring, while in *Drosophila*, both intestinal stem cells and CySCs have higher global translation), a

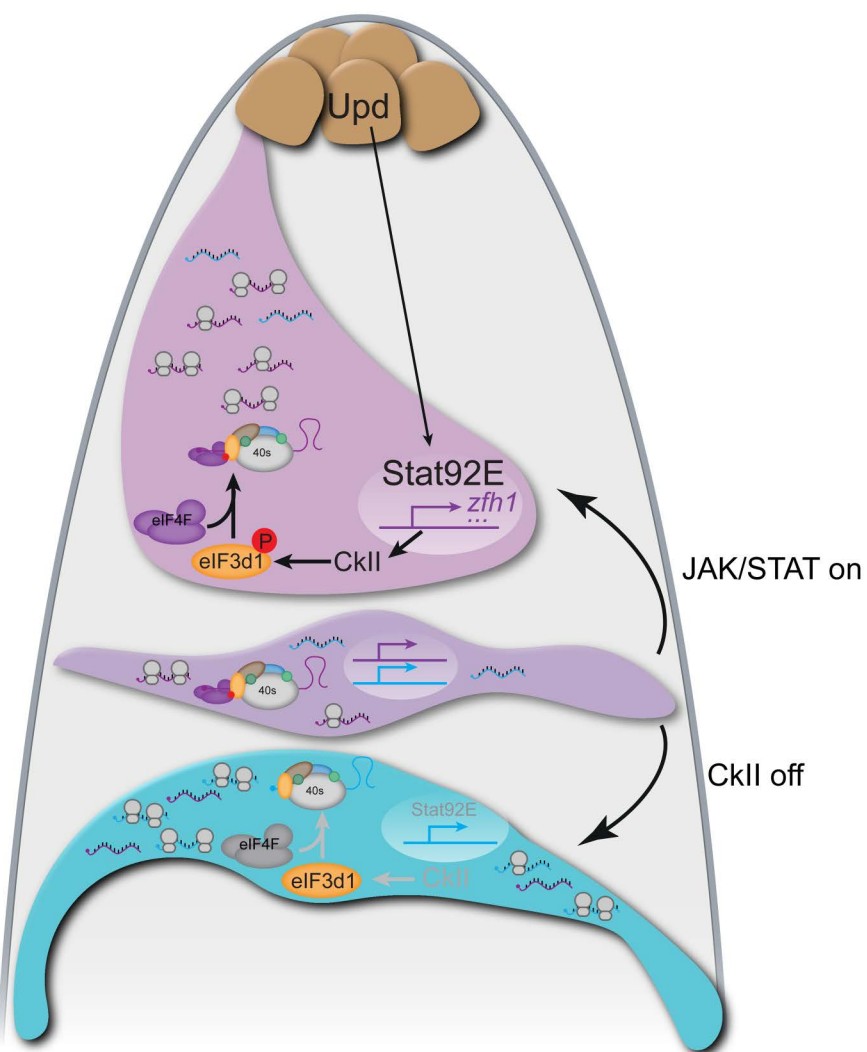

**Fig 9. Model of translational control of stem cell fate in CySCs.** The hub (brown) produces Upd which leads to the stabilization and nuclear translocation of Stat92E in CySCs (magenta). Stat92E drives transcription of genes encoding self-renewal factors (schematized as purple lines), including *zfh1*. Additionally, Stat92E promotes the activity of CkII, at least in part by regulating expression of CkII subunits. In turn, CkII phosphorylates the translation initiation factor eIF3d1 (orange), which promotes the interaction between eIF3d1 and eIF4F, leading to canonical cap-dependent translation. Our data suggest that eIF4F and eIF3d1 together preferentially translate mRNAs encoding self-renewal factors (purple), while mRNAs encoding differentiation factors (cyan), although present in CySCs, are not translated. In CySCs away from the hub, reduced Stat92E allows the transcription of genes encoding both self-renewal factors (purple) and differentiation factors (cyan), but continued CkII activity ensures that translation is restricted to mRNAs encoding self-renewal factors. We suggest these cells are plastic and can self-renew if exposed to JAK/STAT signaling from the niche, or differentiate in response to CkII inactivation. In differentiating cells (blue), Stat92E is not found in the nucleus and CkII is inactive, such that eIF3d1 is unphosphorylated. In this situation, we propose that eIF3d1 no longer interacts with eIF4F, allowing cells to translate mRNAs independently of eIF4F. This results in changed specificity for the mRNAs that are bound to ribosomes, and preferential translation of differentiation factors instead of self-renewal factors.

common feature appears to be that translation is lower when cells finally become post-mitotic [12,16,59–61]. This suggests that the lower requirement for eIF4F activity may be conserved across differentiated cells in multiple tissues. Several studies have found that modulators of eIF4F activity, in particular 4E-BPs, can influence stem cell fate, although most have found

that eIF4F activity is lower in stem cells [62–64]. Intriguingly, DAP5 (also called Novel APOBEC1 target 1 (Nat1) or eIF4G2), which interacts with eIF4A and eIF3, but lacks eIF4E binding and therefore cannot mediate cap-dependent translation, is required for differentiation of ES cells, both in mouse and humans, suggesting that translation of eIF4F-independent transcripts is required for differentiation [65, 66]. Nonetheless, no study to date had evaluated the importance of different initiation factor complexes in stem cell fate determination, and our findings suggest that eIF4F may play specific roles in stem cell self-renewal, while it is dispensable during differentiation.

An important question is how different initiation factors provide specificity in translation. There is growing evidence that initiation factors are not just a passive machinery that translate all mRNAs, but that instead they have preferences either for the sequences around the cap, or for particular ternary structures in the mRNA [67]. Additionally, although there are several mechanisms that are known to allow for translation in the absence of eIF4F [68], our data showing that knockdown of other initiation factors leads to ectopic self-renewal are harder to explain, given how crucial those factors are thought to be in all modes of translation initiation. One possibility is that by using RNAi, we only knocked down gene function incompletely. Another is that these disruptions lead to the activation of cellular stress pathways; there is evidence that activation of stress signaling can promote ectopic self-renewal [69–71]. Thus, unlike the clear effect of loss-of-function of eIF4F, it is still to be determined whether other translation factors drive specific gene expression programmes or whether they are simply required for all translation and lead to non-specific effects upon depletion.

## eIF3d enables a translational switch in response to environmental signals

Our work highlights the role of eIF3d1, the *Drosophila* homologue of eIF3d, as a regulatory switch controlling different translation modes. Knockdown of eIF3d1 only moderately reduced the global translation rate in CySCs (Fig 2I), similar to previous observations in the *Drosophila* embryo [72]. Notably; however, in cell culture, eIF3d knockdown results in one of the more severe reduction in translation rates compared to knockdown of other eIF3 subunits, as measured by ratios of polysomes to monosomes [73]. This may reflect different dependence on eIF3d for translation according to cell types, or may be a result of inefficient RNAi in *Drosophila*. Nonetheless, in several contexts including CySCs, eIF3d plays critical roles in the regulation of gene expression, with dramatic effects on cell proliferation [57,73].

Importantly, eIF3d is subject to regulation by CK2 [28], providing a means for environmental signals such as signals from the stem cell niche to influence translation. Although we did not directly assess the phosphorylation status of eIF3d1 in CySCs in vivo, due to technical limitations, our data strongly suggest that eIF3d1 is indeed phosphorylated by CkII in the testis. We show that phospho-dead eIF3d1, but not a form of eIF3d1 that lacks the ability to bind the mRNA 5′ cap, acts as a dominant-negative for CySC self-renewal, and moreover, that only full length or phospho-mimetic eIF3d1 can rescue self-renewal upon knockdown of CkII. Given that eIF3d1 knockdown phenocopies knockdown of eIF4F components, these data suggest that eIF3d could mediate the interaction between the eIF3-containing 43S ribosome complex to mRNA-bound eIF4F, in a manner that depends on phosphorylation of eIF3d1. In this model, phosphorylation of eIF3d1 would act as a switch to selectively engage eIF4F-dependent translation. Indeed, our experiments using human cells support this hypothesis, showing that CK2 inhibition weakens the interaction between eIF3d and eIF4F. Previous work has shown regulation of eIF3d through phosphorylation, demonstrating that upon glucose deprivation in cells, eIF3d is dephosphorylated and can bind the mRNA cap

directly [28]. However, that work did not assess the interaction between eIF4F and eIF3d as it used a paradigm in which eIF4F activity was independently inhibited, as most studies examining cap-binding roles of eIF3d [74]. In support of our model, studies have established that eIF3d and eIF4G can directly interact and that the presence of eIF3d is essential for the interaction between the core eIF3 octamer complex and eIF4G [54–56]. Thus, we suggest that eIF3d1 phosphorylation is a key regulatory step in the binding of eIF3 and eIF4F, suggesting a novel way in which translation initiation can be regulated by environmental factors through CK2.

## Relative importance of control of global translation rates and targeted translation of specific transcripts

Changes in global translation rates during differentiation have been noted in several stem cell lineages, although often, including germ cells in *Drosophila* or hematopoietic cells in mouse, translation rates increase as stem cells differentiate [16,59,75], contrary to our observations in CySCs in the *Drosophila* testis. Supporting the idea that global rates are important, manipulation of translation rates can influence cell fate. For instance, the loss of long-term self-renewal ability of *pten* mutant hematopoietic stem cells is suppressed by reducing in translation rates, while conversely preventing translation increases by manipulating Tor activity in both male and female germ cells prevents their differentiation [75–77]. Interestingly, high levels of ribosome biogenesis are required in GSCs where translation rates are lower, and reduced ribosome biogenesis results in failed GSC abscission and defective differentiation [75]. Recent work has shown that restoring the balance between ribosome biogenesis and global translation rates is sufficient to allow differentiation [76]; however, the defect observed upon inhibiting ribosome biogenesis depends on the specific translation of *Non1*, which is regulated by a terminal oligo-pyrimidine (TOP) motif in its 5′ UTR [78]. Since Tor also affects the translation of TOP-containing mRNAs, it is possible that it is not global translation rates that are important during GSC differentiation, but the specific translation of TOP-containing mRNAs such as that encoding Non1. Similarly, in ES cells, regulation of eIF4F activity by 4E-BPs has little effect on global translation rates but affects specific transcripts [64,79].

Indeed, our measurements of OPP incorporation in various knockdowns with opposite effects on cell fate suggest that absolute levels of protein synthesis do not drive cell fate decisions (Fig 2I and 2J). Thus, the importance of changing the initiation mode does not lie in the effect on global translation, rather, the mode of initiation must also determine which mRNAs are translated. The female germline illustrates this principle particularly well, as it is increasingly apparent that progression through several developmental stages involves translational control of the genes required for differentiation [14,15]. While much of the literature has focused on specific translational inhibitors binding mRNAs with sequence specificity, it is still unclear how this sequence-specific translational regulation relates to global changes in translation rates of the cell. Our findings in the gonadal soma suggest that changes in global rates via "generic" translation initiation factors do have highly specific effects on gene expression. We identify *eya* as one transcript that is selectively repressed in CySCs, but how this is achieved is not yet clear. Our data suggest a role for eIF3d1, together with eIF4F, in promoting selective translation in CySCs. In some situations, eIF3d has been shown to directly control translation or repression of specific transcripts [25–27,57,72], depending on its association with specific cofactors or by binding to different stem loop structures in the mRNAs. Whether such stem loop structures are present in the *eya* mRNA is yet to be determined. However, in the testis, the role of eIF3d1 appears to be similar to that of eIF4F, suggesting that it acts to maintain eIF4F activity. Thus, specific sequence or structure recognition by eIF3d may be less relevant

to CySC fate decisions and it may be that eIF4F is responsible for driving specificity. Alternatively, it has been suggested that the binding of eIF3d to internal structures within mRNAs could sterically inhibit eIF4F from associating with the 5′ cap [24]. Therefore, it is possible that phosphorylation of eIF3d inhibits its ability to bind RNA at the expense of eIF4F, and that the effects on eIF4F activity are indirect. Determining how phosphorylation affects both canonical and non-canonical aspects of eIF3d activity will be important to understand how it can change the translational programme of cells.

## Distinct translational programmes: Buffering transcriptional errors or enabling cell plasticity?

An important question that arises from our work is why there is a need to control gene expression translationally in CySCs, in addition to the transcriptional control derived from the niche signals. One possibility is that translational regulation provides a buffering mechanism to the cell, ensuring that noise in transcription does not influence cell fate. Indeed, we find transcripts encoding the differentiation factor *eya* present in CySCs when the protein is absent, suggesting that there is a need for such buffering in the *Drosophila* testis. In other tissues, there is extensive evidence from ribosomal profiling experiments that only a fraction of the transcriptome is translated [9–13].

Beyond just acting as a buffer for transcriptional noise, an intriguing hypothesis is that decoupling transcription and translation in this way could explain more generally the difference between specification and commitment during developmental trajectories. During specification, cells acquire a fate but are still labile and can change fate if the environment instructs them to do so [80]. How such plasticity is acquired is still extremely poorly understood; we speculate that overlaying a translational regulation programme onto transcriptional regulation of developmental gene expression could be a critical mechanism to allow cells to acquire a fate transcriptionally, but only commit to it when a second signal instructs a translational change that leads to a new proteome being synthesized and irreversible commitment to a particular differentiated fate (Fig 9). Of note, although specification is best understood in the context of early embryo development, there is evidence that adult stem cells are specified prior to differentiation, albeit under different names ("priming" or "licensing"), both in the *Drosophila* testis and in mammalian tissues [2, 3]. Thus, it is tempting to speculate that decoupling transcription and transcription and allowing CySCs to become specified without committing to differentiation is of critical importance in allowing CySCs to coordinate their differentiation with the germline. CySCs must be licensed to differentiate, yet licensed CySCs do not lose the ability to self-renew if they do not encounter a gonialblast [2]; allowing them to co-express genes involved both in self-renewal and differentiation while ensuring that only one set of genes is translated through a translational programme could ensure both plasticity of cell fate and rapid response to a new environmental signal, such as the presence of a gonialblast, without the need to transcribe new genes prior to beginning differentiation.

Here, we link the self-renewal signal from the niche and translational control via eIF3d1 and eIF4F. This provides an important link to explain how stem cells and differentiating progeny can have different levels of translation or specific translational programmes. While previous work has shown in several stem cell types that translation is selectively regulated, there has been little indication to date as to how signals from the niche could alter translation to influence fate. We suggest a model that could apply to many other situations, and indeed beyond stem cell biology, to cancer, where translation, and increasingly eIF3d, have been shown to sustain cell proliferation and survival [24].

## Materials and methods

### Fly husbandry

Somatic-specific overexpression and knockdown experiments were carried out by using the *tj-Gal4; Tub>Gal80^{ts}* system (referred to as *tj^{ts}*) to allow temporal control of target gene expression in adult flies [81]. Crosses were raised at 18 °C which is permissive for Gal80 activity. Males were collected 0–3 days after eclosion and shifted to the restrictive temperature of 29 °C. Unless otherwise described in the text, flies were kept at 29 °C for 10 days.

For clonal analyses, flies were raised at 25 °C. Adult flies were collected 1–3 days after eclosion and heat shocked in a water bath at 37 °C for 1 h. Clonal CySCs were identified as labeled Zfh1-positive cells adjacent to the hub while clonal cyst cells were identified as either Zfh1-positive cells in the 2nd row from the hub or Eya-positive cells.

For all experiments, *Stat92E^{ts}* refers to the allelic combination *Stat92E^{Frankenstein}/Stat92E^{397}*. Recombinant chromosomes carrying *eIF4E* or *eIF3d1* mutations together with an FRT site were generated by crossing the relevant stocks and screening for resistance to neomycin and lethality.

We used RNAi lines from the Vienna Drosophila Resource Center (VDRC) from the GD and KK libraries, as well as the Kyoto National Institute of Genetics (NIG), all encoding dsRNA constructs and from the Transgenic RNAi Project (TRiP) collection obtained from the Bloomington Drosophila Stock Center (BDSC) which encode shRNAi constructs. We used the following stocks: *UAS-eIF3-S5(eIF3f) RNAi* (VDRC 101465)*, UAS-eIF3-S9 (eIF3b) RNAi* (VDRC 107829), *UAS-eIF3-S10 (eIF3a) RNAi* (VDRC 28140), *UAS-eIF3-p40 (eIF3h) RNAi* (VDRC 106189), *UAS-eIF3-S8 (eIF3c) RNAi* (VDRC 26667), *UAS-eIF3-S6 (eIF3e) RNAi* (VDRC 27032), *UAS-eIF3-S4 (eIF3g) RNAi* (VDRC 28937), *UAS-eIF3-S2 (eIF3i) RNAi* (VDRC 27032), *UAS-eIF4A RNAi* (VDRC 100310), *UAS-eIF4E RNAi* (VDRC 17581), *UAS-eIF4G RNAi* (VDRC 17003)*, UAS-eIF2α RNAi* (VDRC 7799)*, UAS-eIF2γ RNAi* (VDRC 39377), *UAS-eIF1 RNAi* (VDRC 29216), *UAS-eIF1A RNAi* (VDRC 26022), *UAS-eIF6 RNAi (VDRC 108094), UAS-pAbp RNAi* (VDRC 22007), *UAS-eIF4A RNAi* (VDRC 42202), *UAS-eIF4B RNAi* (VDRC 31364), *UAS-eIF4E1 RNAi* (VDRC 7800), *UAS-CKIIα RNAi (2)* (VDRC 330507), *UAS-CKIIβ RNAi (2)* (VDRC 32377), *UAS-eIF3-S9 (eIF3b) RNAi* (BDSC 32880), *ImpL2-GFP* (BDSC 59778), *eIF4E1^{S05891}* (BDSC 8648), *UAS-nls-LacZ* (BDSC 3955), *UAS-nls-LacZ* (BDSC 3956), *UAS-p35* (BDSC 5072), *UAS-p35* (BDSC 5073), *UAS-eIF3-p66 (eIF3d1) RNAi* (NIG 073-09), *eIF3d1^{EY05735}* (BDSC 20072), *eIF3d1^{EP-654a}* (BDSC 43437), *UAS-CKIIα RNAi (1)* (BDSC 35136), *UAS-CKIIβ RNAi (1)* (BDSC 34939), *UAS-CKIIα* (BDSC 24625), *Oregon-R, Stat92e^{Frankenstein}, Stat92e^{397}, tj-Gal4, tub-Gal80^{ts}, y w hsflp122; +; +, ywhsflp^{122} Tub>-Gal4 UAS-nlsGFP; Tub>Gal80 FRT^{40A}, FRT^{40A}, ywhsflp^{122} Tub>Gal4 UAS-nlsGFP;; FRT^{82B} Tub>Gal80, FRT^{82B} ry^{506}, ywhsflp^{122}; ubi-GFP FRT^{40A}, ywhsflp^{122}; ubi-GFP FRT^{80B}, ubi-GFP FRT^{80B}, UAS-upd* (Gift of Erika Bach), *eIF4A^{1013}* (Gift of T. Xie), *eIF4A^{1006}*, (Gift of T. Xie), *UAS-eIF3d1* (Gift of S. Rumpf), *UAS-eIF3d1^{helix11}* (Gift of S. Rumpf), *UAS-eIF3d1^{WT}* (Generated in this study), *UAS-eIF3d1^{DD}* (Generated in this study), *UAS-eIF3d1^{NN}* (Generated in this study). The full list of genotypes used for each figure is shown in S2 Table.

### Generation of transgenic lines carrying eIF3d overexpression constructs

We used the Group-based Prediction System server [82] to predict the phosphorylation sites in *Drosophila* eIF3d1, and compared to the human eIF3d phosphorylation sites reported in [28] on Uniprot. DNA sequences encoding eIF3d^{WT}, eIF3d^{DD} and eIF3d^{NN} were synthesized by Invitrogen GeneArt and cloned into pUAST-attB vector (RRID: DGRC_1419). Cloned plasmids were then injected by BestGene Inc into embryos from a strain carrying the *P{[+t7.7]=CaryP}Msp300[attP40]* landing site and inserted into the genome using PhiC31-mediated recombination.

## Immunohistochemistry

Dissected fly abdomens were fixed in 4% paraformaldehyde in PBS for 15 min then were washed twice in PBS, 0.5% Triton X-100 for 30 min for permeabilization. Permeabilized samples were blocked in PBS, 1% BSA, 0.2% Triton X-100 (PBTB) for 1 h and incubated overnight in primary antibodies diluted in PBTB. Samples were washed twice in PBTB for 30 min each, and incubated in secondary antibodies diluted in PBTB for 2 h at room temperature, followed by washes in PBS, 0.2% Triton X-100. Testes were separated from abdomens and mounted on slides with Vectashield mounting medium for imaging. We used the following antibodies: chicken anti-GFP (1:500, Aves Labs), rabbit anti-GFP (1:500, Invitrogen), rabbit anti-Stat92E (1:500, gift of E. Bach), guinea pig anti-Tj (1:3000, gift of D. Godt), rabbit anti-Zfh1 (1:5000, gift of R. Lehmann), guinea pig anti-Zfh1 (1:5000, this study). Mouse anti-Eya (eya10H6, 1:20, deposited by S. Benzer/N.M. Bonini), mouse anti-Fas3 (7G10, 1:20, deposited by C. Goodman), rat anti-CadN (1:20), rat anti-De-cad (1:20), rat anti-Vasa (1:20, deposited by A.C. Spradling/D. Williams) and mouse anti-Dlg (4F3, 1:20, deposited by C. Goodman) were obtained from the Developmental Studies Hybridoma Bank created by the NICHD of the NIH and maintained at The University of Iowa.

The guinea pig Zfh1 antibody was generated by GenScript. Recombinant antigen consisting of amino acids 648–775 of Zfh1 isoform PB was produced with an N-terminal His-tag used for purification. This antigen was injected into two guinea pigs. The resulting serum was purified by antigen affinity column to obtain concentrated antiserum.

For EdU or OPP staining, abdomens were dissected in Schneider's medium instead of PBS and incubated for 30 min at room temperature in Schneider's medium containing 10 μM EdU or 5 μM OPP. Samples were then fixed, permeabilized and stained with primary and secondary antibodies as above. Click reaction was carried out for 30 min at room temperature in the following reaction buffer: 2.5 μM Alexa picolyl azide (Click Chemistry Tools), 0.1 mM THPTA, 2 mM sodium ascorbate and 1 mM CuSO4.

## In situ hybridization chain reaction (HCR)

HCR was carried out as previously described in [83]. Twenty pairs of probes were designed, tiled along the *eya* transcripts and excluding regions of high similarity with other genes, with initiator sequences corresponding to amplifier B3 for amplification [84]. Probes were purchased from ThermoFisher as DNA oligos, sequences are listed in S3 Table. Adult abdomens were dissected, fixed, and washed with PBS, 0.5% Triton X-100 as detailed above. Afterward, samples were incubated in Probe Hybridization Buffer (Molecular Instruments) for 30 min at 37 °C in water and incubated with pre-mixed probe pairs (0.01 μM for each probe) at 37 °C overnight. Abdomens were then washed four times for 15 min each with Probe Wash Buffer at 37 °C followed by a wash for 10 min with 5× saline sodium citrate solution (SSCT): 14.61 g/mol sodium chloride, 73.53 g/mol 560 sodium citrate, pH 7, with 0.001% Tween 20 at room temperature. Samples were then incubated with Amplification Buffer for 10 min at room temperature. Meanwhile, 12 pmol of hairpin H1 and H2 were snap-cooled (heat at 95 °C for 90 s and cooled to room temperature for 20 min) separately to prevent oligomerisation. The snap-cooled hairpins were then added to the samples in the Amplification Buffer and incubated overnight at room temperature. On the following day, samples were washed with 5× SSCT for 10 min then incubated with DAPI for 2h. After washing with ×1 PBS for 30 min, testes were mounted on slides as above.

## Imaging and quantification

Images were acquired with a Zeiss LSM880 or LSM800 confocal microscope and analyzed using ImageJ. CySC numbers were assessed by counting all Zfh1-positive, Eya-negative cells. To measure OPP fluorescence intensity, individual cyst cells were outlined using Dlg staining. All

cells that could be unequivocally identified as CySCs, using contact with the hub as a marker, were counted in each sample. To account for variability in OPP between samples, measurements were normalized to either the hub or to GSCs, depending on whether the manipulations were likely to also affect those cell types (e.g., leaky expression of tj-Gal4 in hub cells, or likely effect of the *Stat92e^{ts}* mutation on GSCs). For Stat92E quantifications, fluorescence in the Stat92E channel was used to outline cells and measurements were normalized to levels in GSCs in the same samples, or, for clonal experiments, to the mean of all neighboring non-mutant CySCs that could be unequivocally identified in the same sample. For in situ hybridization, individual cells were harder to outline, so signal was measured in a single plane crossing the center of the hub for each sample, using an ROI defined as the ImpL2-positive region contacting the hub for CySCs, and ImpL2-positive regions away from the hub for cyst cells.

## Cell culture and immunoprecipitations

$2 \times 10^6$ HeLa cells stably expressing Flag-tagged eIF3d or eIF3d$^{DD}$ were seeded in 10 cm petri dish overnight. For the negative control, $1.5 \times 10^6$ HeLa cells were seeded in 10 cm dish two days prior to the experiment, followed by transfection of Flag-RAP2A on the next day with the use of Lipofectamine 2000 (Thermo Fisher, #12566014). On the following day, cells were treated for 2 h either with 3 μM CX-5011 (AOB1816-5, Aobious) or DMSO, followed by quick wash with PBS and lysis with 400 μM of lysis buffer (50 mM Tris pH7.5, 150 mM NaCl, 1% Triton-X supplemented with 2× protease inhibitor cocktail (Roche, 11836145001), 1× phosphatase inhibitor cocktail (Roche, 11836170001), Sodium fluoride (50 mM), glycerol 2-phosphate (1 g/l), Sodium vanadate (2 mM), and Benzonase (50 U/ml)). Protein concentration of lysates was estimated with Pierce BCA (Life technologies, 23224, 23228). Based on the observed concentrations samples were balanced, a 10th of the samples was set aside and mixed with 5× Laemmli buffer to serve as the input control. The rest of the sample was mixed with anti-flag-beads (Sigma, A2220-5 ml), which were prewashed three times with lysis buffer. The sample was incubated with the beads for 2 h, rotating at 4 °C. To remove unbound fraction, the sample was washed three times with 500 μl of lysis buffer. Prior to the last wash the beads were moved into a new tube. Elution was performed by boiling the beads with 100 μl of Laemmli solution. The samples were further analyzed by conventional western blot assay. We used the following antibodies: anti-eIF4G1 (Cell Signaling #8701), anti-eIF4A (Cell Signaling #2490), anti-eIF3b (Santa Cruz, discontinued), phospho-CK2 Substrate (pS/pT) (Cell Signaling #8738), anti-FLAG (Sigma, F7425), all at 1:1000.

## Statistical analysis

We used GraphPad Prism software to analyze data and generate graphs. For statistical analysis, we either used non-parametric ANOVA (Kruskal-Wallis test) followed by Dunn's multiple comparisons test, to compare CySC numbers when there were several genotypes to compare, or Mann–Whitney tests for direct comparisons of CySC numbers when there were only two conditions. For normalized OPP level comparison, we used either parametric ANOVA (Šidák multiple comparisons test) or Students *t* test. Chi-squared tests were applied for categorical data. Statistical tests used in each experiment are indicated in the relevant figure legends.

## Supporting information

**S1 Table. Summary of initiation factor screen.** Percent of testes with ectopic CySCs (assessed as Zfh1-positive, Eya-negative cells at least 3 cell diameters from the hub), or fewer than 10 CySCs near the hub. Note that some knockdowns with ectopic CySCs also occasionally led to a complete absence of somatic cells, so when ectopic cells were observed, the reduction of CySCs near the hub was not assessed.

(DOCX)

**S2 Table.  List of genotypes used in each figure.**
(DOCX)

**S3 Table.  List of probes used for in situ hybridization chain reaction.**
(XLSX)

**S1 Fig.  OPP incorporation depends on proximity to the niche rather than cell fate.** (**A**) A testis in which *Rbf* was knocked down by RNAi, resulting in ectopic CySCs away from the hub and absence of differentiated cyst cells (marked with Eya, yellow). OPP incorporation (white) shows that the global translation rate in CySCs adjacent to the hub (arrows) is higher than in ectopic CySCs (arrowheads). Zfh1 (magenta) labels CySCs and Dlg (yellow) labels cell outlines. Asterisks indicate the hub. Scale bar: 15 μM. (**B**) Quantification of OPP incorporation in niche CySCs and ectopic CySCs, normalized to the hub ($N \geq 67$ cells from 9 testes, Student $t$ test, ****$P < 0.0001$). Underlying data for all graphs can be found in file <u>S1 Data</u>.
(TIF)

**S2 Fig.  eIF4F is specifically required for CySC self-renewal** (**A–D**) Testes with positively marked control clones (A,B) or clones homozygous for *eIF4A$^{1013}$* (C,D) at 2 dpci (A,C) and 7 dpci (B,D). GFP (cyan) labels the clone. Control clones are readily recovered in CySCs at 2 dpci and maintained at 7 dpci, while mutant clones are rarely observed adjacent to the hub. Zfh1 (magenta) labels CySCs and Eya (yellow) labels differentiated cells. Arrows mark CySCs and arrowheads mark differentiated cells. Asterisks indicate the hub. Scale bar: 15 μM. (**E**) Percentage of testes with positively-marked control or *eIF4A* mutant clones in either CySCs or differentiated cyst cells at 2 dpci ($N \geq 27$ testes, Chi-squared test, ns $P = 0.3443$). (**F**) Percentage of testes with positively-marked control or *eIF4A* mutant CySC clones at 2 dpci and 7 dpci. ($N \geq 27$ testes, Chi-squared test, **** $P < 0.0001$). (**G**) Percentage of testes with negatively-marked control or *eIF4A* mutant clones in either CySCs or differentiated cyst cells at 2 dpci ($N \geq 30$ testes, Chi-squared test, ns $P = 0.3517$). (**H**) Percentage of testes with negatively-marked control or *eIF4A* mutant CySC clones at 2 dpci and 7 dpci. ($N \geq 30$ testes, Chi-squared test, **** $P < 0.0001$). (**I–J′**) Testes with negatively-marked control clones (E, E′) and clones homozygous mutant for *eIF4A$^{1013}$* (F, F′) at 7 dpci. Clones are identified by lack of GFP (cyan). Mutant CySC clones are not recovered. Zfh1 (magenta) labels CySCs and Eya (yellow) labels differentiated cells. Arrows mark CySCs and arrowheads mark differentiated cells. Asterisks indicate the hub. Scale bar: 15 μM. (**K–L′**) Testes with negatively-marked control clones (G, G′) and clones homozygous mutant for *eIF4E1$^{S058911}$* (H, H′) at 7 dpci. Clones are identified by lack of GFP (cyan). Mutant CySC clones are not recovered. Zfh1 (magenta) labels CySCs and Eya (yellow) labels differentiated cells. Arrows mark CySCs and arrowheads mark differentiated cells. Asterisks indicate the hub. Scale bar: 15 μM. (**M**) Percentage of testes with negatively-marked control or *eIF4E1* mutant clones in either CySCs or differentiated cyst cells at 2 dpci ($N \geq 24$ testes, Chi-squared test, ns $P = 0.3327$). (**N**) Percentage of testes with negatively-marked control or *eIF4E1* mutant CySC clones at 2 dpci and 7 dpci ($N \geq 24$ testes, Chi-squared test, ***$P < 0.001$). Underlying data for all graphs can be found in file <u>S1 Data</u>.
(TIF)

**S3 Fig.  eIF4F-deficient CySCs are not lost by apoptosis** (**A**, **B**) A control (*tj$^{ts}$*>+) testis (A) and a testis in which eIF4G1 was knocked down for 7 days (B), labeled with antibodies against the activated caspase, Dcp-1 (yellow, single channel A′,B′) to mark apoptotic cells, Tj (magenta) to label CySCs and early cyst cells, and Eya and Fas3 (cyan) to label late-stage cyst cells and the hub respectively. No increase in Dcp-1-positive cells (arrows) is visible. Scale bar: 15 μM. (**C**, **D**) Dcp-1 expression in control (*tj$^{ts}$*>LacZ; LacZ) testis (C) and a testis in which

the baculovirus caspase inhibitor was over-expressed in cyst cells ($tj^{ts}$>*LacZ; P35*) (D). Dcp-1 positive cells (arrows) are rarely observed in testes with P35 over-expression compared to the control. Tj (magenta) labels early cyst cells, Eya and Fas3 (cyan) labels late stage cyst cells and the hub respectively, and Dcp-1 (yellow) labels cells undergoing apoptosis. Scale bar: 15 µM. (**E**) Percentage of testes with Dcp-1-positive cells in control and testes in which P35 was over-expressed in the cyst lineage ($N \geq 18$ testes, Chi-squared test, **** $P < 0.0001$). (**F**) Number of Zfh1+ Eya⁻ CySCs in testes in which the indicated initiation factors were knocked down together with inhibition of apoptosis by co-expression of P35. Blocking cell death did not rescue CySC loss caused by knockdown of initiation factors ($N \geq 15$ testes, Kruskal–Wallis test, ****$P < 0.0001$, ns $P > 0.9999$). Underlying data for all graphs can be found in file S1 Data. (TIF)

**S4 Fig. eIF4A mutant cyst cells associate with germ cells** (A–C) Testes with membrane-labeled control clones (A, A′) and two examples of clones homozygous mutant for *eIF4A¹⁰¹³* (B–C′) at 5 dpci. RFP (cyan) labels the clones (arrows). Both control and mutant clones display the characteristic morphology of cyst cells, with a long flattened cytoplasm enveloping germ cell cysts (arrows). Tj (magenta) labels cyst cells and Vasa (yellow) labels germ cells. Scale bar: 15 µM. (TIF)

**S5 Fig. Knockdown of initiation factors results in ectopic CySC-like cells.** (**A–C‴**) A control ($tj^{ts}$> +) testis (A–A‴), and testes in which eIF2α (B–B‴) or eIF3a was knocked down in CySCs (C–C‴). EdU (cyan) labels cells in S phase. In controls, EdU-positive somatic cells are found only adjacent to the hub, while in eIF knockdowns, EdU-positive CySC-like cells are observed away from the hub. Zfh1 (magenta) labels CySCs and Eya (yellow) labels differentiated cyst cells. Arrows mark CySCs, dashed arrows indicate ectopic Zfh1-positive cells away from the hub, and arrowheads mark differentiated cells. Asterisks indicate the hub. Scale bar: 15 µM. (**D**) Percentage of testes with Edu+ Zfh1+ cells at least 2 cell diameters from the hub ($N \geq 15$ testes, Chi-squared test, ****$P < 0.0001$, **$P < 0.01$). Underlying data for all graphs can be found in file S1 Data. (TIF)

**S6 Fig. eIF4F and eIF3d1 act downstream of JAK/STAT signaling in CySCs.** (**A, B**) Expression of the JAK/STAT target *Socs36E* detected by in situ hybridization chain reaction (cyan, single channel in A′,B′) in a control ($tj^{ts}$> +) testis (A), and a testis in which eIF4A (B) was knocked down in CySCs, after 2 days at 29 °C. No apparent change in *Soc36E* mRNAs is visible. Zfh1 (magenta) labels CySCs. Asterisks indicate the hub. Scale bar: 15 µM. (**C**) Quantification of *Socs36E* fluorescence signal in CySCs around the hub in the indicated genotypes, normalized to the levels in the hub ($N \geq 20$ testes, Šidák multiple comparisons test, ns $P > 0.9$). (**D**) A testis in which Upd was over-expressed with concomitant knockdown of eIF4A in CySCs ($tj^{ts} > eIF4A RNAi, upd$). N-cadherin (yellow) labels the hub. Zfh1 (magenta) labels CySCs and Eya (cyan) labels differentiated cyst cells. Arrows mark CySCs and arrowheads mark differentiated cells. Asterisks indicate the hub. Scale bar: 15 µM. (**E**) Percentage of testes with Eya-positive, Zfh1-negative differentiating cyst cells in the indicated genotypes ($N \geq 15$ testes, Chi-squared test, ****$P < 0.0001$). (**F**) Detection of cell death in a testis in which eIF3d1 was knocked down in CySCs for 7 days with an antibody against activated Dcp-1 (yellow). Dcp-1-positive cells are rarely observed. Tj (magenta) labels early cyst cells, Eya and Fas3 (cyan) label cyst cells and the hub respectively. Scale bar: 15 µM. (**G**) Quantification of the number of Zfh1⁺ Eya⁻ CySCs in control or eIF3d1 knockdown testes, with or without inhibition of apoptosis using the baculovirus caspase inhibitor P35. Blocking cell death did not rescue CySC numbers upon eIF3d1 knockdown ($N \geq 15$ testes, Kruskal–Wallis test,

****$P < 0.0001$, ns $P > 0.9999$). (**H**) Expression of the JAK/STAT target *Socs36E* detected by in situ hybridization chain reaction (cyan, single channel in H′) in a testis in which eIF3d1 was knocked down in CySCs, after 2 days at 29 °C. Zfh1 (magenta) labels CySCs. Asterisks indicate the hub. Scale bar: 15 μM. (**I**–**J**) Testes in which Upd was over-expressed with concomitant knockdown of eIF3d1 in CySCs (*tj$^{ts}$ > eIF3d1 RNAi, upd*). Eya-positive cells were present in all samples examined, but some still showed ectopic Zfh1-positive CySCs (I, 10/17 testes), while in others Zfh1-positive cells were mostly absent (J, 7/17 testes). N-cadherin (yellow) labels the hub. Zfh1 (magenta) labels CySCs and Eya (cyan) labels differentiated cyst cells. Arrows mark CySCs and arrowheads mark differentiated cells. Asterisks indicate the hub. Scale bar: 15 μM. Underlying data for all graphs can be found in file S1 Data.
(TIF)

**S7 Fig. eIF3d1 phosphorylation but not cap-binding is required for CySC self-renewal.** (**A**) Top: alignment of the N terminus of human eIF3d and *Drosophila* eIF3d1. Asterisks indicate conserved residues, colons indicate residues with strongly similar properties and periods indicate residues with weakly similar properties. The two serines subject to phosphorylation in human eIF3d are highlighted in red boxes along with predicted phosphorylation sites in *Drosophila*. Bottom: predicted phosphorylation sites in *Drosophila* eIF3d1 using Group-based Prediction System (GPS) server. The boxed lines indicate serines 536 and 539, highlighted above, which are strong predicted targets of CkII. (**B**) Number of Zfh1$^+$ Eya$^-$ CySCs in testes in which wild-type (eIF3d1$^{WT}$), phospho-mimetic (eIF3d1$^{DD}$) or phospho-dead (eIF3d1$^{NN}$) eIF3d1 was over-expressed. Phospho-dead eIF3d1 expression results in a significant decrease in CySC numbers ($N \geq 18$ testes, *$P < 0.05$, ns $P = 0.1150$ (eIF3d1$^{WT}$) or $P > 0.9999$ (eIF3d1$^{DD}$), Kruskal-Wallis test). (**C**) Quantification of OPP levels in CySCs in indicated genotypes, normalized to the GSCs ($N \geq 62$ cells from $\geq 8$ testes, Šidák multiple comparisons test, ns $P = 0.275$, *$P = 0.023$). (**D**) Number of Zfh1$^+$ Eya$^-$ CySCs in testes in which wild-type eIF3d1 or a form of eIF3d1 in which mutations are introduced in the mRNA 5′ m$^7$G cap-binding domain (eIF-3d1$^{helix11}$) were over-expressed ($N \geq 15$ testes, Kruskal–Wallis test, ns $P > 0.9999$). (**E**) Western blot from lysates of cells expressing a control construct (RAP2A) or eIF3d and labeled with an antibody against a pan-phospho-CK2 substrate (left) or stained for total protein (right). Addition of the CK2 inhibitor CX-5011 resulted in decreased labeling for the phospho-CK2 substrate. Underlying data for all graphs can be found in file S1 Data, raw images of western blots can be found in file S1 Raw Images.
(TIF)

**S8 Fig. CkII knockdown does not affect JAK/STAT signal transduction.** (**A**, **B**) A control testis (A) and a testis in which CkIIβ was knocked down in CySCs (B), stained with an antibody against Stat92E (yellow, single channel A′,B′) after 20h at the restrictive temperature. Tj (magenta) labels CySCs and early cyst cells, Dlg (cyan) labels the hub and Eya (cyan) labels differentiated cyst cells. Arrows mark CySCs. Asterisks indicate the hub. Scale bar: 15 μM. (**C**) Quantification of Stat92E levels in CySCs in control testes or upon CkIIβ knockdown, normalized to levels in neighboring GSCs. ($N > 100$ cells from $\geq 7$ testes, Student t *test*, ns $P = 0.5057$) (*D*) Expression of the JAK/STAT target *Socs36E* detected by in situ hybridization chain reaction (cyan, single channel in D′) in a testis in which CkIIα was knocked down in CySCs for 2 days. Zfh1 (magenta) labels CySCs. Asterisks indicate the hub. Scale bar: 15 μM. (**E**) Quantification of *Socs36E* fluorescence signal in CySCs around the hub, normalized to the levels in the hub ($N \geq 16$ testes, Šidák multiple comparisons test, ns P = 0.06). Underlying data for all graphs can be found in file S1 Data.
(TIF)

**S1 Data. Underlying numerical data for graphs in Figs 1H–1J, 2H–2J, 4F, 4G, 5C, 5F, 5I, 6D–6F, 6I, 6J, 7C, 7F, 7I, 8B, 8E, S1B, S2E–S2H, S2M, S2N, S3E, S3F, S5D, S6C, S6E, S6G, S7B–S7D, S8C, S8E.**
(XLSX)

**S1 Raw Images. Images of complete blots and explanation of how they were processed to obtain the images shown in Figs 6G, 6H and S7E.**
(PDF)

## Acknowledgments

We thank S. Rumpf, E. Bach, T. Xie, R. Lehmann and D. Godt and the Bloomington, Vienna and Kyoto *Drosophila* stock centres for fly stocks and reagents. The authors are grateful to Ivana Bjedov, Nazif Alic, Richard Poole, Deepika Vasudevan, Vilaiwan Fernandes and members of the Amoyel and Fernandes labs for critical discussions and comments on the manuscript.

## Author contributions

**Conceptualization:** Ruoxu Wang, Aurelio A. Teleman, Marc Amoyel.

**Data curation:** Marc Amoyel.

**Funding acquisition:** Marc Amoyel.

**Investigation:** Ruoxu Wang, Mykola Roiuk, Freya Storer.

**Methodology:** Ruoxu Wang, Mykola Roiuk.

**Project administration:** Marc Amoyel.

**Supervision:** Aurelio A. Teleman, Marc Amoyel.

**Writing – original draft:** Ruoxu Wang, Marc Amoyel.

**Writing – review & editing:** Ruoxu Wang, Marc Amoyel.

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
