## [Editor Report · Decision Letter 0]

2 Feb 2024

Dear Dr Amoyel, 

Thank you for submitting your manuscript entitled "Niche signalling regulates eIF3d1 phosphorylation to promote distinct modes of translation initiation in stem and differentiating cells" for consideration as a Research Article by PLOS Biology.

Your manuscript has now been evaluated by the PLOS Biology editorial staff and I am writing to let you know that we would like to send your submission out for external peer review.

Once your full submission is complete, your paper will undergo a series of checks in preparation for peer review. After your manuscript has passed the checks it will be sent out for review. To provide the metadata for your submission, please Login to Editorial Manager (https://www.editorialmanager.com/pbiology) within two working days, i.e. by Feb 04 2024 11:59PM.

Kind regards,

Lucas Smith

Senior Editor

PLOS Biology

---

## [Decision Letter · Decision Letter 1]

5 Apr 2024

Dear Dr Amoyel,

Thank you for your patience while your manuscript "Niche signalling regulates eIF3d1 phosphorylation to promote distinct modes of translation initiation in stem and differentiating cells" was peer-reviewed at PLOS Biology as a Research Article. Please accept my sincere apologies for the delays that you have experienced during the peer review process. Your manuscript has been evaluated by the PLOS Biology editors, an Academic Editor with relevant expertise, and by four independent reviewers.

As you will see in the reviewer reports, the reviewers are generally positive about your study and think it is potentially interesting and well done. However, they raise a substantial number of important and overlapping concerns regarding the preliminary nature of the mechanistic insights underpinning the translational control mechanism. Specifically, Reviewer #1 raises concerns that the study lacks direct evidence for eIF3d1 phosphorylation and asks that additional data using phosphomimetic mutants is included to demonstrate that the eIF3D-eIF4G1 interaction is CK2-dependent. The reviewer also suggests demonstrating the importance of eIF3d1 phosphorylation in another Drosophila stem cell type to help generalise the findings. In addition, several reviewers note that the findings from key experiments conducted in HeLa cells should be replicated in Drosophila cells.

Based on their specific comments and following discussion with the Academic Editor, it is clear that a substantial amount of work would be required to meet the criteria for publication in PLOS Biology. However, given our and the reviewer interest in your study, we would be open to inviting a comprehensive revision of the study that thoroughly addresses all the reviewers' comments. Given the extent of revision that would be needed, we cannot make a decision about publication until we have seen the revised manuscript and your response to the reviewers' comments. We would also be willing to review a revision plan document before you start working on the revision due to the amount of suggested experiments. Your revised manuscript would need to be seen by the reviewers again, but please note that we would not engage them unless their main concerns have been addressed. 

We appreciate that these requests represent a great deal of extra work, and we are willing to relax our standard revision time to allow you 6 months to revise your study. Please email us (plosbiology@plos.org) if you have any questions or concerns, or envision needing a (short) extension.

**IMPORTANT - SUBMITTING YOUR REVISION**

*Resubmission Checklist*

*Published Peer Review*

*PLOS Data Policy*

*Blot and Gel Data Policy*

Sincerely,

Richard

Richard Hodge, PhD

rhodge@plos.org

REVIEWS:

Reviewer #1: In this manuscript, Wang and colleagues report a biological phenomenon in which mRNA translation mediated by eIF4F-eIF3d1 is crucial for the self-renewal of somatic cyst stem cells (CySCs) of the Drosophila testes. Mechanistically, they demonstrate the importance of eIF3d1 phosphorylation by CK2, caused by the activation of JAK-STAT ligands from the testis niche. Importantly, eIF4F was found to be dispensable in differentiating cells, but is only required for the self-renewal of CySCs. Collectively, the authors identify a physiological role for translational control in determining stem cell fate. Although the finding does appear to demonstrate the importance of translational control in an in vivo setting, there is a stark lack of mechanistic details underlying the proposed specialized translation. Moreover, CK2-mediated mobilization of the eIF4F complex or phosphorylation-dependent control of eIF3 are already reported phenomena, and therefore lacks novelty. Hence, the reviewer finds that the manuscript in its current form requires major revisions to be suitable for publication in PLoS Biology.

Major Comments:

1. In the text describing Figure 3A and B, authors claim that JAK/STAT signaling remains functional in mutant cells and that eIF4F knockdown does not greatly interfere with the translation of JAK/STAT signaling components, which are overreaching conclusions based solely on the presence/absence of Stat92E alone. To demonstrate this in greater detail, the authors should look for phospho-JAK or the relative expression of Stat92E target genes. 

2. The authors have only demonstrated association of human eIF3D with eIF4F, which is already known as the authors mention in the text, in Figure 5G using HeLa cells. However, as the study is centered around Drosophila genetics, the authors should assess the interaction of Drosophila eIF3d1 with eIF4F by expressing them in human cells, or perform endogenous immunoprecipitation using S2 cells.

3. The authors simply "infer" that eIF3d1 is phosphorylated, by using phosphomimetic mutants and based on previous reports that CK2 can phosphorylate eIF3, but do not provide any direct solid evidence (i.e. phospho-antibody/orthophosphate labeling) to conclude that eIF3d1 is truly undergoing phosphorylation-dependent control. This is especially important here, since the authors are claiming that CK2 phosphorylation-mediated effects on eIF3D cap-binding activity are not important under this setting. 

4. The decrease in binding between eIF3D and eIF4G1in Figure 5H is not very convincing, despite having quantified the difference in Figure 5I. To clarify the result the authors should perform either a dose/time course with the CK2 inhibitor, or assess the interaction following genetic depletion of CK2. Importantly, to firmly conclude that the eIF3D-eIF4G1 interaction is CK2-dependent, the authors should test whether the inhibitor has no effect on the interaction between eIF4G1 and eIF3D phosphomimetic mutant. Also worth noting is that the authors used a very broad CK2 inhibitor for their experiments, which has more than 100 substrates including other subunits of eIF3. 

5. The molecular link between JAK-STAT and CK2 remains missing. What do the authors hypothesize is happening? Although it has been reported that CK2 phosphorylates and activates JAK and therefore downstream JAK-STAT signaling, it does not explain how the expression of CKIIa rescues the phenotype of Stat92E mutant flies shown in Figure 6I. Are there other targets of CK2 that act to promote JAK-STAT activation? These ideas should be discussed in the Discussion. 

6. The mechanism behind how differentiation genes (i.e. eya) are selectively translationally repressed in CySCs is completely absent in this manuscript. While it may be out of the scope of this manuscript to systematically profile the ribosome-protected fragments (RPFs), the authors should discuss the possible underlying reasons for this in the Discussion. For example, do there exist stem loop structures within the 5'UTR of eya that may display reduced affinity towards eIF3d1 binding so that it is rarely translated in CySCs? 

7. Even though the authors focused on the CySCs as their model system, it would be interesting to assess the importance of eIF3d1 phosphorylation-mediated translational control in another Drosophila stem cell type to assess the universality of this regulation, again since the phosphorylation-dependent cap-binding switch does not appear to be important here. Could this finding then be generalized as a species-specific event? 

8. The authors' finding that eIF3d1 is crucial for global translation conflicts with a previous study (Szostak et al. Hrp48 and eIF3d contribute to msl-2 mRNA translational repression . Nucleic Acids Research, 2018; see Figure 5E) which showed only a "mild defect" in global translation upon eIF3d1 knockdown using Drosophila embryo extracts. The authors should compare and contrast the findings, and whether it could be attributed to the difference in cell types (i.e. embryo vs. testis)

Minor Comments:

1. For the RNAi experiments, there are no control experiments showing the degree or efficiency of knockdown, for example by Western blot or FISH, making it difficult to conclude whether the observed phenotypes can be directly comparable. 

2. The quantification strategy for immunohistochemistry data needs to be better explained in the Methods section, as the figures themselves do not always seem to reflect the dramatic changes illustrated on the quantified graphs. 

3. There are numerous typos in the manuscript that deter readability. The authors should go through the text again carefully, also making sure that the Figures/Figure legends align properly with their descriptions within the Results. 

Reviewer #2: This is an interesting paper that describes how the dynamic regulation of mRNA translation plays roles in both self-renewal and differentiation of cyst stem cells (CyCSs) within the Drosophila testis. OPP pulse labeling reveals subtle differences in global protein production between CySCs, their immediate daughters and their further differentiated progeny. RNAi knockdown experiments suggest eIF4F plays a specific role in CySCs, while the reduction of other tested translation factors resulted in the accumulation of ecotpic Zfh1 positive cells. The authors then present data that indicates eiF4F is epistatic to JAK/STAT. Interestingly, eiF4d appears to play a specific role in CySC self-renewal. Further work shows that CKII promotes the interaction between eIF4d and eIF4F. The paper ends with experiments showing that CKII mediates both autonomous and non-autonomous effects of niche signaling in CySCs.

While the ideas and results presented in the manuscript are interesting and worth pursuing, several weaknesses significantly dampened enthusiasm for the paper. Overall the results appear preliminary and should be further developed before publication in PLoS Biology.

Major:

The authors conclude that eIF4F knockdown results in specific effects of CySC self-renewal that are different from other translation factors. However no controls/data are presented for the expression of these genes within the CySC lineage and after RNAi knockdown. Are these phenotypes rescuable by non-targeted transgenes?

Unclear how the OPP immunofluorescence was quantified. Were whole cells used or regions of interest within the selected cells? OPP labeling can be variable from sample to sample, so care should be taken when evaluating results.

eIF4F and the other translation factors are all essential genes, and it seems unlikely that severe knockdown would allow cells to survive. This should be quantified. A slightly amended model to the one being proposed by the authors is that CySCs produce and contain all the protein needed for differentiation before their daughters begin to move away from the niche.

The authors rule out cell death as a possible explanation for the loss of cells based on negative results from p35 over-expression. No controls for the effectiveness of p35 in suppressing cell death in this system are given. In addition, no assays to detect cell death in its many forms are presented. The authors should provide data on these points.

The authors present data on the effects of different eIF4d transgenes, but the expression of each transgene in the experiments should be evaluated before firm conclusions can be made.

The authors propose a provocative model whereby eIF4d activity results in the translation of specific messages. However, the identity of those messages remains unknown and thus the model is untested. It is unclear whether any target mRNAs show changes in translation through direct regulation by these factors.

The authors should block CySC lineage differentiation independent from expanding JAK/STAT signaling, if possible and evaluate OPP labeling. Experiments like this one is critical for testing the idea that niche signaling and not differentiation per se regulates protein synthesis within this lineage.

Minor:

Do manipulations expected to modulate protein synthesis (mTOR activation etc) result in changes in OPP labeling?

Several groups have investigated dynamic mRNA translation in the female germline, but few if any of these references are cited or discussed.

Reviewer #3: This manuscript by Wang, et al. provides an interesting and novel perspective on how the stem cell niche microenvironment contributes to stem cell fate regulation through influencing translational control. The goal of this study was to identify and describe a mechanism of translational regulation using somatic cyst stem cells (CySCs) and their differentiated cyst cell progeny from the well-studied Drosophila male testis. The authors studied translation initiation factors and found different requirements for translation initiation cap binding proteins in CySCs versus differentiated cyst cells. They dissect this mechanism further, pinpointing a phosphorylation event that regulates the interaction between cap binding proteins that promotes different translation programs. This gives cells the ability to rapidly respond to environmental signals by switching the translation mode. Altogether, this study adds an interesting perspective on how translational control provides a tunable response mechanism for protein expression that influences cell fate decisions. The findings are relevant to many biological situations that require rapid adaptation including development, tissue homeostasis, and diseased states.

The experimental strategies described in this study are well-designed, utilize orthogonal approaches and appropriate controls, and are presented logically throughout this well-written manuscript. The authors further provide adequate background information at appropriate places such that a non-expert can understand and appreciate the topic. Despite this, there are a few questions and addressing these would help clarify some aspects. 

Experimental or data interpretation considerations: 

1. I am curious as to how a general shift in temperature shift translational rates. If the data in Fig 1H were to be plotted normalized to the WT control in Fig 1B, what would that look like? 

2. The model in Fig. 7 indicates a steep change of JAK-STAT in CySC vs. cyst cell with the Stat92E to be ON in CySC but entirely OFF in cyst cell, but immunostaining using anti-Stat92E in the control in Fig. 3A does not seem to show this pattern?

3. How does the JAK-STAT signaling control the activity of the CKII kinase activity or expression level or both?

4. Immunoprecipitations were done using Hela cells, how can these cells recapitulate the protein-protein interactions in the somatic gonadal cells in fly testis? 

Minor notes:

1. Please include the microscope and software used for imaging in the methods. 

2. Additional details on quantification of OPP should be present in the methods:

a. Was this done in Fiji or Imaris?

b. Are these measurements done on a single Z-slice or from the entire cell?

c. What was used for the normalization? (it was mentioned in the figure legend, not the methods that the hub was used) 

3. Double check that the stock information for all fly lines used in the study are in the methods. Fly lines used for the clonal analysis seem to be missing:

a. Example: Supplemental Figure 1P shows RFP clones. What is the membrane bound protein that is tagged with RFP to indicate these clones? This seems to be a different fly line than what was used to make the eiF4A clones in the same figure. There, both the positive and negative clones have a nuclear signal.

4. To a non-expert, the localization pattern of ImpL2-GFP is unclear in Figure 2L-2O. Is it cytoplasmic, membrane-bound, nuclear?

a. Is the ImpL4-GFP a transgenic or knock-in fly line?

b. The signal is quite weak, thus indicating ImpL2's localization within CySCs by drawing the boundaries of a single cell would be helpful.

c. A zoom-in of a single cell will clarify true eya RNA signal looks like. 

5. For Figures 4A and 5A-C, is there an additional (grey) channel that is not labelled?

6. VDRC RNAi: Are these dsRNA or shRNA? 

Reviewer #4: In this study, the authors have described an original model to elucidate how cell identity transitions occur in Drosophila testis cyst stem cells. Initially, they demonstrate that global translation rates are high in CySCs and decrease during differentiation in a JAK/STAT-dependent manner. However, by knocking down different translation initiation factors, they show that the global rate of translation does not determine cell fate. Instead, they identify two initiation factors, eIF4F and eIF3d1, as crucial for CySCs' self-renewal and fate determination.

Indeed, the authors find that eIF4F and eIF3d1 are necessary for CySCs' self-renewal. They observe that eIF4F acts downstream of JAK/STAT activity, and they demonstrate that Casein Kinase II phosphorylation of eIF3d1 is crucial in this pathway. Overall, they suggest that CySCs have a specific translational program that requires the eIF4F complex regulated by eIF3d1 phosphorylation. According to the proposed model, stem cells possess mRNAs encoding self-renewal and differentiation factors that can be selectively translated by switching a translational program, thereby determining cell fate.

Overall, there is much to commend in this manuscript. Initial knock-down experiments using RNAi are supported by genetic mutants for the most significant genes, using both positively and negatively marked genetic clones. The loss of stem cells in specific mutant conditions is demonstrated to result from differentiation rather than cell death (p35 experiments). Moreover, genetic experiments and interactions are nicely complemented by molecular experiments showing that the phosphorylation of eIF3d1 could impact its molecular partners. Thus, this manuscript extends beyond describing a genetic pathway important for stem cell self-renewal and differentiation. Furthermore, the manuscript is candid in describing unexpected results that contradict the initial assumptions of the authors.

However, several crucial points need addressing to validate some of their results and test the authors' model. Specifically, the authors place translational regulation at the core of their model to switch between self-renewal and differentiation. Ideally, one would over-express an RNA sufficient for differentiation in stem cells with its endogenous translational regulatory regions (5' and 3' UTR). Their model predicts that this should not be translated and thus should not lead to stem cell differentiation. Then, repeating the same experiment with ubiquitous 5' and 3' UTR should result in translation and in stem cell loss. Alternatively, the authors could conduct the reverse experiment with an RNA sufficient for self-renewal and express it with and without its UTRs in differentiating somatic cells.

Major comments:

1) When testing the importance of translational control, some key results are presented in Figure 2L-O, where the authors compare the expression of eya mRNA and Eya protein. However, the images lack clarity, making it difficult to discern cell locations, especially in close-up captions. For instance, in Figure 2M-M', eya mRNA is detected in both CySCs (dashed arrow) and differentiating cells (arrowhead), while Eya protein is only observed in differentiating cells (Figure 2L' arrowhead). Nevertheless, the levels of eya mRNA appear markedly different between stem cells (low) and differentiating cells (high), raising doubts about whether the difference in protein levels might partly result from varying levels of transcription. To address this concern, it would be important to overexpress eya mRNA in stem cells and observe any resulting appearance of Eya protein.

2) The model posits a switch of translation factors that determine cell fate. It would be important to show, under wild-type conditions, the endogenous expression of key eIF factors through immunofluorescence or tagged proteins. Additionally, it would be instructive to investigate the behavior of these eIF factors in the presence of JAK/STAT mutations and during upd upregulation to determine if their expression profiles and localization align with the model proposed in this manuscript.

3) In Fig 3A,B, and B'': how do the authors select cells to normalize fluorescence? In 3B'', a cell on the right side is chosen (arrowhead), but why not select the one on the left side with higher expression?

4) The authors show that eIF4F functions downstream of JAK/STAT in CySC self-renewal. In Figure 3E, it is observed that eIF4G1 mutation rescues the upd overexpression. Do other eIF4 factors also rescue the phenotype? Can an eIF3d mutation rescue upd overexpression too?

5) In Fig. 5F, the expression levels of wt, NN, and DD eIF3d could differ in the testis, potentially explaining the varying rescue profiles. Do they exhibit similar expression levels?

6) The experiments depicted in Figure 5 are conducted in human HeLa cells. Is eIF3d phosphorylated in Drosophila testis?

7) It would be interesting to test whether overexpression of eIF4F could rescue ckII mutants, suggesting that critical levels of eIF4F are crucial for self-renewal, independent of eIF3d phosphorylation.

8) In Fig. 6, the authors demonstrate that overexpression of CKIIα can rescue stat92 mutants. Can the expression of phosphomimetic eIF3d rescue stat92 mutants compared to phosphor-dead or wild-type versions?

9) In Figure 2B, it is unclear whether the different mutants are sterile.

Minor comments:

1. In the introduction, authors say "… used as a marker for CySCs (Inaba et al. 2011)…". This article focuses on String and cdc25, it could be replaced by Tulina Matsunis Science 2001 for example? 

2. In the introduction, authors explain the model and mention Fig1A as support. This figure needs labels for CySCs, GSCs, 1st row and 2nd row to better understand the rest of the figures. 

3. In the introduction, authors explain translation initiation without a Figure Support. The scheme of Fig.2A could be included in FIG.1 for better understanding of the model they suggest from the beginning. 

4. In the introduction, authors could consider to mention Ghosh Lasko Plos One 2015 about eIF4E, eIF4E-3, eIF4G and eIF4G2 during spermatogenesis, and Shao. et al. Developement 2023, about eIF4E5 which is important for spermatogenesis. 

5. The quantifications are explained in each legend and only sometimes in the main text. They should be explained systematically also in the main text since the differences are sometimes subtle and without the quantification technique is less convincing. 

6. Fig.1G': the light violet dashed 2nd row lines are difficult to observe. 

7. Results: Overexpression of upd results in higher levels of OPP incorporation in Zfh1 positive cells (Fig.1I): Could zfh1 impact ribosome biogenesis explaining the increase of translation? 

8. Results Fig S1G-J, the label eiF4E1 is difficult to see. 

9. Results Fig S1Q, Vasa perinuclear staining seems affected compared to the control. What happens in eiF4E? 

10. Results Fig.2GH, CySCs continue to self-renew, is this independent of Jak/Stat? Is it possible that eiF3H inhibits zfh1 transcription? 

11. Authors write " …phospho-mimetic eIF3d1 had no effect on CySC numbers (Fig.S3B)": this is wrong, it should be S3C. 

12. Authors write "However, expression of this construct did not affect CySC numbers (Fig.S3C)". This is wrong, it should be S3D. 

13. Authors write "…with a pan phosphor CK2 substrate antibody (Fig.S3D)": it should be S3B.

14. In the graphical abstract/working model (Fig.7), homogenize with the scheme of Fig1 by labelling the important cells of the study directly in the figure: CySCs, 1st row, 2nd row, etc.

---

## [Decision Letter · Decision Letter 2]

17 Jan 2025

Dear Marc,

Thank you for your continued patience while we considered your revised manuscript "Niche signalling promotes distinct modes of translation initiation in stem and differentiating cells via regulation of eIF3d by CK2." for publication as a Research Article at PLOS Biology. Please accept my sincere apologies for the delays that you have experienced during this round of the peer review process. This revised version of your manuscript has been evaluated by the PLOS Biology editors, the Academic Editor and the three of the original reviewers.

Based on the reviews, I am pleased to say that we are likely to accept this manuscript for publication, provided you satisfactorily address the remaining points raised by the reviewers. After discussions with the Academic Editor, we will not make the additional molecular data requested by Reviewer #1 on eIF3D phosphorylation necessary for the revision, but we do ask that the requested textual and presentational changes are incorporated.

In addition, please make sure to address the following data and other policy-related requests that I have provided below (A-F):

(A) We routinely suggest changes to titles to ensure maximum accessibility for a broad, non-specialist readership. In this case, we would suggest a minor edit to the title, as follows. Please ensure you change both the manuscript file and the online submission system, as they need to match for final acceptance:

"Signals from the niche promote distinct modes of translation initiation to control stem cell differentiation and renewal in Drosophila"

(B) You may be aware of the PLOS Data Policy, which requires that all data be made available without restriction: http://journals.plos.org/plosbiology/s/data-availability. For more information, please also see this editorial: http://dx.doi.org/10.1371/journal.pbio.1001797

-Supplementary files (e.g., excel). Please ensure that all data files are uploaded as 'Supporting Information' and are invariably referred to (in the manuscript, figure legends, and the Description field when uploading your files) using the following format verbatim: S1 Data, S2 Data, etc. Multiple panels of a single or even several figures can be included as multiple sheets in one excel file that is saved using exactly the following convention: S1_Data.xlsx (using an underscore).

-Deposition in a publicly available repository. Please also provide the accession code or a reviewer link so that we may view your data before publication. 

Figure 1H-J, 2H-J, 4F-G, 5C, 5F, 5I, 6D-F, 6I-J, 7C, 7F, 7I, 8B, 8E, S1B, S2E-H, S2M-N, S3E-F, S5D, S6C, S6E, S6G, S7B-D, S8C, S8E

(C) Please also ensure that each of the relevant figure legends in your manuscript include information on *WHERE THE UNDERLYING DATA CAN BE FOUND*, and ensure your supplemental data file/s has a legend.

(D) We require the original, uncropped and minimally adjusted images supporting all blot and gel results reported in the following Figures:

Figure 6G-H, S7E

We will require these files before a manuscript can be accepted so please prepare and upload them now. Please carefully read our guidelines for how to prepare and upload this data: https://journals.plos.org/plosbiology/s/figures#loc-blot-and-gel-reporting-requirements

(E) Please ensure that your Data Statement in the submission system accurately describes where your data can be found and is in final format, as it will be published as written there. 

(F) Per journal policy, if you have generated any custom code during the course of this investigation, please make it available without restrictions. Please ensure that the code is sufficiently well documented and reusable, and that your Data Statement in the Editorial Manager submission system accurately describes where your code can be found. 

We expect to receive your revised manuscript within three weeks. 

*Published Peer Review History*

*Press*

Best wishes,

Richard

Richard Hodge, PhD

rhodge@plos.org

Reviewer remarks:

Reviewer #1: I appreciate the author made useful improvements to the manuscript. 

Two major outstanding issues remain:

1. It seems very important to show that eIF3d is phosphorylated since the phosphorylation state and eIF4F interaction is one of the major findings of the manuscript. Can authors use proteomics to detect modification or at least a proxy of CK2-eIF3d interaction? 

2. It is still difficult to understand how a very subtle ~10-25% reduction (depending on TBB or CX-5011 treatment) in non-phosphorylated eIF3d-eIF4F interaction leads to changes in CySC self renewal. 

The expression of the endogenous UTR-containing Eya construct is a helpful addition. Can the authors could use a variation of this experiment to at least prove that this small change leads to gene-specific changes by expressing a heterologous UTR-containing Eya construct and showing this is now resistant to CKII inhibition and changes to eIF3d phosphorylation? This would allow authors to better link a small shift in interaction to specific translation control. 

In addition, in Figure 6G, equal eIF4A is being pulled down with the RAP2A control as with eIF3d WT. This is also seen in Figure R1 with RAP2A versus eIF3D DD for eIF4A1, since the IP is not cleaner it is difficult to consider the influence of eIF3d phosphorylation of the eIF4F and eIF3D interaction.

Minor comment:

Authors state: "We show that phospho-dead eIF3d1, but not a form of eIF3d1 that lacks the mRNA 5' cap- binding domain, acts as a dominant-negative for CySC self-renewal, and moreover, that only full length or phospho-mimetic eIF3d1 can rescue self-renewal upon knockdown of CkII."

I could not find this data with eIF3d1 without the 5' cap binding domain - does the author mean a mutant where the cap-binding function is blocked (eIF3d1-helix11)? If so this is also incorrectly described in the results. 

Reviewer #3: The authors have done an overall good job for the revision and addressed most of the previous questions. However, I do have an issue for the FISH data shown in the new Figure 8: Given the high background in particular in the area containing cyst cells, it is really hard to tell whether these are real signals or which cells these "dots" belong to. In many cases, it looks like from the germ cells instead of cyst cells to me. The data quality in Figure 3 using similar approach gave out better images. I think this needs to be addressed, in particular if these are representative images used for quantification. 

Reviewer #4: The revised version of the manuscript by Wang and colleagues has greatly improved. We congratulate the authors for their constructive answers to our comments and for performing additional important experiments. We don't have any additional comment, except praising the authors for this nice work.

---

## [Editor Report · Decision Letter 3]

3 Feb 2025

Dear Marc,

On behalf of my colleagues and the Academic Editor, Wendy Gilbert, I am pleased to say that we can accept your manuscript for publication, provided you address any remaining formatting and reporting issues. These will be detailed in an email you should receive within 2-3 business days from our colleagues in the journal operations team; no action is required from you until then. Please note that we will not be able to formally accept your manuscript and schedule it for publication until you have completed any requested changes.

PRESS

Best wishes, 

Richard

Richard Hodge, PhD

rhodge@plos.org

PLOS
